# Global NO and HONO emissions of biological soil crusts estimated by a process-based non-vascular vegetation model

Philipp Porada[1,2], Alexandra Tamm[2], Jose Raggio[3], Yafang Cheng[2], Axel Kleidon[4], Ulrich Pöschl[2], and Bettina Weber[2]

[1]University of Potsdam, Vegetation Ecology and Nature Conservation, Am Mühlenberg 3, 14476 Potsdam, Germany
[2]Max Planck Institute for Chemistry, P.O. Box 3060, 55020 Mainz, Germany
[3]Departamento de Biología Vegetal II, Facultad de Farmacia, Universidad Complutense de Madrid, Madrid, Spain
[4]Max Planck Institute for Biogeochemistry, P.O. Box 10 01 64, 07701 Jena, Germany

**Correspondence:** Philipp Porada (philporada@uni-potsdam.de)

**Abstract.** The reactive trace gases nitric oxide (NO) and nitrous acid (HONO) are crucial for chemical processes in the atmosphere, including the formation of ozone and OH radicals, oxidation of pollutants and atmospheric self-cleaning. Recently, empirical studies showed that biological soil crusts are able to emit large amounts of NO and HONO and they may therefore play an important role in the global budget of these trace gases. However, the upscaling of local estimates to the global scale is subject to large uncertainties, due to unknown spatial distribution of crust types and their dynamic metabolic activity. Here, we perform an alternative estimate of global NO and HONO emissions by biological soil crusts, using a process-based modelling approach to these organisms, combined with global datasets of climate and land cover. We thereby consider that NO and HONO are emitted in strongly different proportions, depending on the type of crust and their dynamic activity, and we provide a first estimate of the global distribution of four different crust types. Based on this, we estimate global total values of $1.04 \, \text{Tg} \, \text{yr}^{-1}$ NO-N and $0.69 \, \text{Tg} \, \text{yr}^{-1}$ HONO-N released by biological soil crusts. This corresponds to around 20 % of global emissions of these trace gases from natural ecosystems. Due to the low number of observations on NO and HONO emissions suitable to validate the model, our estimates are still relatively uncertain. However, they are consistent with the amount estimated by the empirical approach, which confirms that biological soil crusts are likely to have a strong impact on global atmospheric chemistry via emissions of NO and HONO.

## 1 Introduction

Biological soil crusts (hereafter "biocrusts") are complex communities of organisms which cover large areas around the globe, mainly in arid and semiarid regions (Sancho et al., 2016). They consist of various different species of free-living cyanobacteria, green algae, lichens, bryophytes, fungi, and bacteria. In contrast to vascular plants, biocrust organisms are able to dry out and then restart their metabolic activity depending on the availability of water, which explains their abundance in regions of extreme climatic conditions. Predecessors of today's biocrusts may have formed the first terrestrial ecosystems in the Proterozoic, 2 billion years ago (Lenton and Daines, 2017). It has been suggested that these early biocrusts were already highly productive and that they may have influenced global atmospheric composition and climate (Porada et al., 2016b; Lenton et al., 2016).

Also today's biocrusts have been suggested to affect biogeochemical cycles, both at the regional and the global scale (Elbert et al., 2012; Sancho et al., 2016). In drylands, they provide several essential ecosystem services, such as protection of the soil surface against erosion (Rodríguez-Caballero et al., 2012; Rodríguez-Caballero et al., in press) and input of carbon, nitrogen and phosphorus into the soil (Barger et al., 2016). Carbon input into the ecosystem is carried out by lichens, bryophytes, cyanobacteria, and algae, which constitute the primary producers of biocrusts (Tamm et al., 2018). Fixation of atmospheric nitrogen is due to cyanobacteria which are either free-living or hosted as symbionts by lichens and bryophytes (Zhao et al., 2010; Rousk et al., 2013). Phosphorus is trapped in form of dust particles by lichens and bryophytes, or may be acquired through enhanced weathering of surface rocks by the organisms (Porada et al., 2014, 2016b). Input of carbon and nitrogen into biocrusts is the precondition for microbial activity, which leads to transformation of organic nitrogen and subsequent release of nitrogen trace gases, such as nitric oxide (NO), nitrous acid (HONO) and nitrous oxide ($N_2O$) (Abed et al., 2013; Barger et al., 2016).

To estimate the global impacts of these biocrust-related biogeochemical processes, empirical upscaling of field measurements has been performed. By extrapolating field observations of productivity grouped into ecosystem classes, Elbert et al. (2012) estimated a global net primary productivity (NPP) of 0.6 Gt yr$^{-1}$ of carbon for biocrusts in desert and steppe ecosystems. This represents around 6 % of total NPP in the world's arid and semiarid regions (Huang et al., 2016) and around 1 % of global terrestrial NPP (Ito, 2011). Moreover, Elbert et al. (2012) used their approach to estimate the importance of biocrusts for biotic nitrogen fixation by natural terrestrial ecosystems. They estimated that biocrusts contribute around 25 % to biotic nitrogen fixation in desert and steppe regions around the world, which corresponds to approximately 12 % globally (Cleveland et al., 1999).

In addition to inputs of nitrogen by biocrusts into terrestrial ecosystems, emissions of different nitrogen species by biocrusts at the global scale have been quantified. Lenhart et al. (2015) estimated global emissions of $N_2O$ by lichens, bryophytes, cyanobacteria, and algae based on large-scale patterns of their NPP as presented by Elbert et al. (2012). The authors found that these organisms are responsible for 4 - 9 % of global emissions of $N_2O$ from natural terrestrial sources. In drylands, where they represent main components of biocrusts, they may even contribute up to 100% to $N_2O$ emissions. This has implications for global climate, since $N_2O$ acts as an efficient greenhouse gas and depletes stratospheric ozone.

Apart from $N_2O$, biocrusts and soils emit further nitrogen trace gases, such as NO and HONO (Su et al., 2011; Oswald et al., 2013; Weber et al., 2015). These gases are crucial for atmospheric chemistry, since they control the formation of OH radicals, which are in turn necessary for the oxidation of atmospheric pollutants and which also affect cloud formation (Seinfeld and Pandis, 2006; Su et al., 2011). Weber et al. (2015) found that biocrusts emit $\sim$1.7 Tg yr$^{-1}$ of nitrogen in the form of NO and HONO at the global scale, which corresponds to $\sim$20 % of global nitrogen oxide emissions from natural ecosystems (Ciais et al., 2013). This points at an important role of biocrusts for global atmospheric processes.

While these studies, which are based on empirical upscaling of field measurements, suggest significant impacts of biocrusts on global biogeochemical cycles, they are usually subject to large uncertainties. The reason for this is the low number and small scale of measurements of biocrust-related biogeochemical functions in the field, which generally show high variation.

As an alternative method to the empirical upscaling of field measurements, process-based models may be applied to quantify global biogeochemical effects of biocrusts. In general, these models predict the functioning of the organisms based on climate and other environmental conditions. Since high-resolution climate data are available at the global scale, the model estimates do not depend on the upscaling of a low number of field measurements to global values.

In contrast to vascular vegetation, however, the non-vascular photoautotrophs which form biocrusts are seldom considered in global process-based vegetation models. An exception to this is the LiBry model, which is specifically designed for non-vascular vegetation. LiBry predicts photosynthesis, respiration, growth and dynamic surface cover of lichens, bryophytes, terrestrial cyanobacteria and algae as a function of climate (Porada et al., 2013). An important aspect of the LiBry model is its explicit representation of functional diversity of vegetation. Instead of aggregating diversity into a low number of functional types, LiBry simulates multiple physiological strategies, similar to the JEDI vegetation model (Pavlick et al., 2013), which increases the realism of the simulated vegetation.

The model has already been used in various studies on the effects of non-vascular vegetation on global biogeochemical cycles. Global NPP of lichens and bryophytes was simulated by LiBry (Porada et al., 2013), and was found to be consistent with the empirical estimate by Elbert et al. (2012). Further applications of the LiBry model include climate effects of early non-vascular vegetation in the geological past (Porada et al., 2016b; Lenton et al., 2016), and also effects on permafrost soil processes at high latitudes (Porada et al., 2016a). Moreover, global $N_2O$ emissions by non-vascular vegetation were estimated by LiBry, based on the simulated respiration of the organisms (Porada et al., 2017).

Global emissions of NO and HONO by biocrusts, however, have not yet been estimated using the LiBry model or another process-based modelling approach. The main reason for this is the strong dependence of observed NO and HONO emissions on the type of the biocrust (Weber et al., 2015). The types are categorised according to the dominant photoautotrophic organisms of the biocrust, such as cyanobacteria, lichens and mosses. Consequently, estimating total NO and HONO emissions by biocrusts for a certain region requires knowledge of the relative abundance of different local photoautotrophs, which has to be considered in the modelling approach. Another complicating factor is the marked nonlinear dependence of NO and HONO emissions on the water saturation of the biocrust (Weber et al., 2015), which therefore has to be simulated at a high temporal resolution. Given the potential large contribution of biocrusts to global NO and HONO emissions, however, a refined estimate in this regard is crucial for assessing the significance of biocrusts for global atmospheric chemistry.

The objective of this study is to provide an alternative estimate of global NO and HONO emissions by biocrusts, based on the process-based non-vascular vegetation model LiBry.

To this end, we extended the LiBry model in three central aspects: First, we introduced a scheme which categorizes the large number of physiological strategies simulated by LiBry for drylands into lichens, mosses and cyanobacteria. We then defined the different biocrust types considered in the study by Weber et al. (2015) according to these vegetation groups. This enabled us to take into account the strong differences in NO and HONO emissions between biocrust types. Secondly, we altered the scheme for dynamic surface cover of the physiological strategies in LiBry, which enabled us to predict the relative cover of each biocrust type. Thirdly, we extended LiBry by an empirical scheme which calculates NO and HONO emissions of different biocrust types based on their water saturation. Thereby, saturation of the biocrusts is based on the dynamic water content of the

individual physiological strategies simulated by LiBry. We evaluated our estimates of biocrust surface cover both at the local and the global scale by comparison to observations and we compared simulated NO and HONO emissions to the available estimates from the literature.

Please note that in the remainder of the manuscript, we will use the term 'moss' instead of 'bryophyte', although the group of bryophytes also includes liverworts and hornworts in addition to mosses. The reason for this terminology is that the corresponding biocrust is usually called 'moss-dominated biocrust'.

## 2 Methods

### 2.1 Description of the LiBry model

The Lichen and Bryophyte model (LiBry) is a process-based vegetation model, which is specifically designed for non-vascular organisms. The original model version, published in Porada et al. (2013), was developed to quantify global NPP of lichens and mosses. Since then, LiBry has been extended in various aspects, which has increased its applicability to questions of global biogeochemistry (Porada et al., 2016b, a, 2017).

LiBry computes photosynthesis and respiration of non-vascular vegetation as a function of climate and other environmental conditions. Photosynthesis is calculated by the Farquhar-scheme (Farquhar and von Caemmerer, 1982), and thus depends on light, $CO_2$, and temperature. To account for the adjustment of metabolism to water availability, which is characteristic for non-vascular vegetation, photosynthesis also depends on the water saturation of the organisms, as simulated by the model. Furthermore, the decreasing effect of water saturation on diffusion of $CO_2$ into non-vascular organisms is considered in the model. Respiration is computed as a function of temperature through a $Q_{10}$-relationship, and it also depends on water saturation, in the same way as photosynthesis. The dynamic water saturation of the organisms is calculated based on the balance of water inputs via rainfall, snow melt, and dew, and water losses due to evaporation. Thereby, evaporation and surface temperature of the organisms are derived from the surface energy balance, using a modified Penman-Monteith equation (Monteith, 1981; Porada et al., 2013). Hence, the climate data which are necessary to drive the model comprise short-wave solar radiation, downwelling long-wave radiation, air temperature, relative humidity, rainfall, snowfall, and near-surface wind speed.

Several physiological properties regulate the dynamic water saturation of non-vascular vegetation in the model: First, the uptake of water is limited not only by rainfall or snow melt, but also by the water storage capacity of the organisms, which depends on it height and the porosity of the biomass. At full saturation, additional water input infiltrates into the soil. The extent to which dew can be used as a water source in the model depends mostly on climatic conditions, and to a limited extent on properties of the organisms which influence the surface temperature, e.g. via evaporation. Secondly, also water loss is regulated by properties of the organisms. These are the same as for water uptake, namely the specific water storage capacity, capillary structure of the biomass and albedo. Note that non-vascular vegetation does not possess stomata, so an active reduction of evaporation is not possible.

NPP is calculated as the difference of photosynthesis and respiration, and translated into growth of biomass, thereby accounting for tissue turnover, which leads to loss of biomass. Growth and loss of biomass lead to a dynamic surface cover in the

model, which further depends on the frequency of disturbances. Therefore, additionally to climatic fields, data on the interval of disturbance events, such as fire, for instance, are required to run the model. Furthermore, leaf and stem area index of vascular vegetation are needed, since the canopy influences radiation and moisture supply to ground-based non-vascular organisms, and it may also serve as habitat for non-vascular epiphytes. Climate data and other boundary conditions are provided as time series for a large number of points on a global grid.

In contrast to most global vegetation models, which aggregate diversity into a few average plant functional types, LiBry simulates a large number of physiological strategies, similar to the JEDI vegetation model (Pavlick et al., 2013). In this way, LiBry explicitly represents functional diversity of non-vascular vegetation. To create the strategies, a Monte-Carlo approach is used, which randomly samples ranges of observed physiological properties derived from many studies on non-vascular organisms (see appendix B in Porada et al. (2013)). For each location on the global grid, only a fraction of all initial simulated physiological strategies will be able to survive under the given climatic conditions. This simulated natural selection leads to global patterns of physiological strategies in the model, which are driven by climate. An advantage of this approach is that the model selects the appropriate parameters for each set of climatic conditions itself, it is thus not necessary to determine the physiological parameters of all species everywhere.

## 2.2 Representation of biocrust types in LiBry

For this study we extended the LiBry model to make possible the calculation of NO and HONO emissions by biocrusts. In a first step, we introduced different types of biocrusts into LiBry. This is motivated by the strong dependence of NO and HONO emissions on the type of biocrust, which was found by Weber et al. (2015). They distinguished four biocrust types, named after the dominating photoautotrophic organisms:

(a) light cyanobacteria-dominated biocrusts, colonised by a thin layer of cyanobacteria

(b) dark cyanobacteria-dominated biocrusts, colonised by a thicker, more dense layer of cyanobacteria, which may grow together with cyanolichens

(c) chlorolichen-dominated biocrusts, colonised by green-algal lichens, and

(d) moss-dominated biocrusts, colonised by mosses.

Furthermore, Weber et al. (2015) found that NO and HONO emissions show a marked, nonlinear relation to the water saturation of the biocrusts. Hence, for our study, we used their full data set to establish average NO and HONO emissions as a function of water saturation and biocrust type (see Fig. 1). Moreover, we assumed a $Q_{10}$-value of 2.0 and a reference temperature of 25° C for the dependence of NO and HONO emissions on ambient temperature (Weber et al., 2015).

To implement these relations into the LiBry model, we discretized the curves shown in Fig. 1 and created a look-up table, which assigns values of NO and HONO emissions for each value of water saturation. Subsequently, the emissions were scaled according to surface temperature:

$$E_{\mathrm{NO,ONO}} = E_{\mathrm{NO,HONO}}(\Theta)Q^{\frac{T_{\mathrm{S}}-T_{\mathrm{REF}}}{10.0}} \tag{1}$$

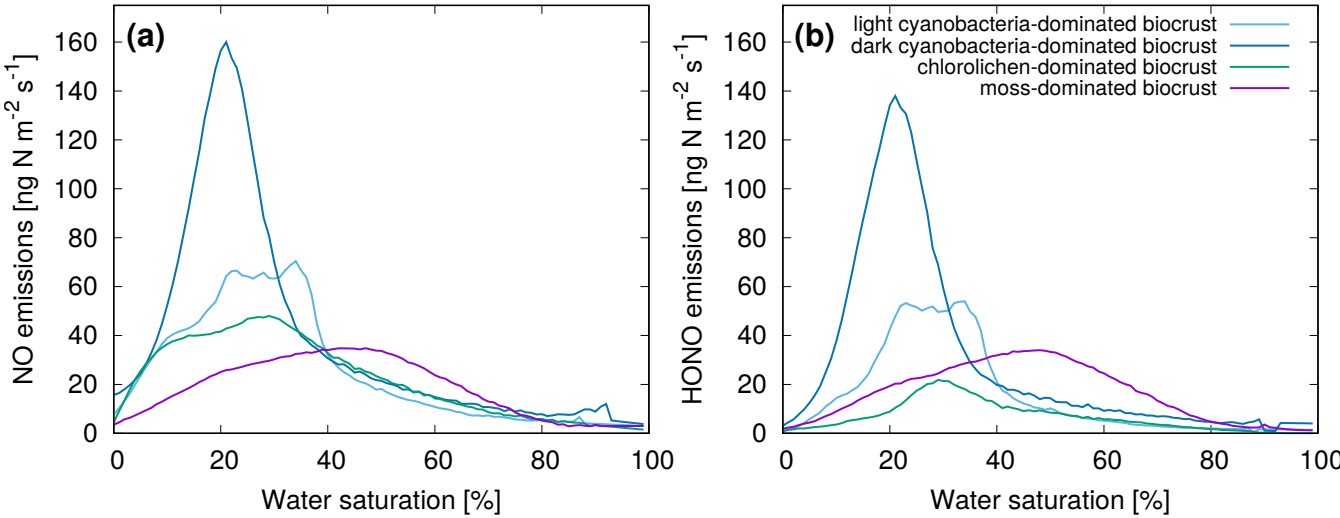

**Figure 1.** Average emissions of (a) NO and (b) HONO as a function of water saturation of four different biocrust types. The emissions were measured on biocrust samples in the laboratory at 25° C. See Weber et al. (2015) for further details.

where $E_{NO,HONO}$ are the emissions of NO and HONO, respectively, $E_{NO,HONO}(\Theta)$ are the emissions at a given water saturation $\Theta$ based on the look-up table, $Q$ is the $Q_{10}$-value, and $T_S - T_{REF}$ is the difference between the surface temperature of the simulated organisms and the reference temperature. In this way, NO and HONO emissions were calculated from the simulated water saturation at each time step of the model run.

It should be mentioned that this approach does not simulate the complete nitrogen cycling in biocrusts. The input of nitrogen, either via fixation or deposition, and losses of nitrogen in form of leaching or gases other than NO and HONO, are not represented. The strong observed NO and HONO emissions by cyanobacteria-dominated biocrusts (Fig. 1) indicate that the nitrogen source may be biotic fixation. However, further quantitative studies are needed to implement these processes in biocrust models.

To represent different biocrust types in LiBry, each of the many physiological strategies simulated by LiBry was assigned to one of the four biocrust types. We want to point out that LiBry simulates only photoautotrophic organisms, not the whole biocrust continuum which additionally includes fungi, soil bacteria and animals, and the mineral soil. However, the dominating photoautotrophic organisms exert a strong influence on the microbial composition and the physiological functioning of the whole biocrust (Maier et al., 2018). This justifies the classification of the four biocrust types listed above according to their

dominant photoautotrophs and, consequently, the utilisation of physiological strategies simulated by LiBry as indicators of the biocrust type. We only considered strategies growing in drylands, since these are the main regions where biocrusts occur at the large scale.

To assign strategies to biocrust types, we first determined to which group of photoautotrophs (lichens, mosses or cyanobacteria) each simulated physiological strategy belonged. This was necessary since the LiBry model does not categorize the

simulated strategies by default. Instead, an individual strategy is defined only through its unique combination of values of

several physiological parameters, as described above. We used these physiological parameters to distinguish the strategies into lichens, mosses and cyanobacteria. For this purpose, the following parameters were taken into account: Height, $CO_2$-diffusivity in the wet state, and photosynthetic capacity. The growth height of a strategy has several effects in the model: For the same amount of cover expansion, the higher a strategy is, the more biomass is needed, which is a competitve disadvantage. However, taller strategies have more potential to store water per given area, and they may also outcompete smaller strategies with regard to light availability. The $CO_2$-diffusivity at high water saturation is an important physiological constraint, since organisms with higher diffusivity are able to grow more than those with low diffusivity in the model. This advantage is, however, associated with increased loss of water through evaporation for given climatic conditions, due to the more open structure of the biomass. Photosynthetic capacity controls the ability of a photoautotroph to use high light intensities and to capture $CO_2$ from the atmosphere. Strategies with a high photosynthetic capacity are able to grow more than those with low capacity under certain climatic conditions, but this advantage comes at the cost of increased maintenance respiration and turnover. We want to mention that the categorization of strategies into lichens, mosses and cyanobacteria has no impact on the dynamics of the vegetation in the model, it only affects the simulated NO and HONO emissions.

We used a stepwise scheme to identify the growth form of a physiological strategy simulated by LiBry (see Fig. 2): If the height of the simulated strategy exceeds 2 mm, it is either a lichen or a moss, otherwise it is a cyanobacterium. It should be clarified that a physiological strategy in LiBry does not correspond to an individual organism, but rather to a layer of biomass with certain physiological properties. The value of 2 mm therefore does not correspond to the size of an individual cyanobacterium, but it represents the assumed upper limit for the height of a layer of cyanobacterial biomass.

To distinguish between lichens and mosses, we used the range of $CO_2$-diffusivity at full water saturation based on all strategies in a global simulation. If the $CO_2$-diffusivity of a strategy lies in the upper half of this range, it is a moss, otherwise it is a lichen, since mosses generally show higher diffusivities at saturation than lichens (Cowan et al., 1992; Williams and Flanagan, 1998). While lichens and mosses in biocrusts may show a certain amount of overlap in their $CO_2$-diffusivity, we are not aware of any systematic studies in this regard. Hence, we applied the general, simple pattern of distinct $CO_2$-diffusivities here, which may be further refined in future studies. We then assumed that lichens indicate chlorolichen-dominated biocrusts and mosses indicate moss-dominated biocrusts.

To differentiate cyanobacteria in the model, we used the photosynthetic capacity of the strategies, which roughly corresponds to their Rubisco-content per area. If the photosynthetic capacity of a less than 2 mm tall strategy lies in the upper half of the simulated global range, the strategy is assumed to indicate dark, and, otherwise, light cyanobacteria-dominated biocrusts. In general, light and dark cyanobacteria-dominated biocrusts differ by their abundance of cyanobacteria per area, while there is no reason why the photosynthetic capacities of the individual organisms should differ. However, the LiBry model represents the organisms as a layer of biomass with a certain Rubisco concentration, which is proportional to photosynthetic capacity. Thus, LiBry actually captures the effect of a lower abundance of cyanobacteria in light crusts compared to dark crusts by simulating a layer with a lower total photosynthetic capacity.

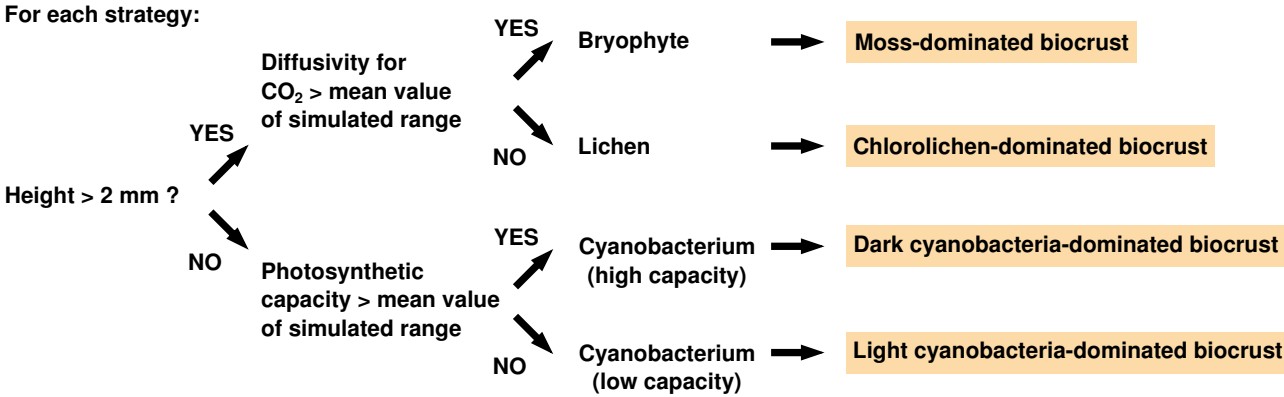

**Figure 2.** Scheme to assign each simulated strategy in the LiBry model to one of four biocrust types for dryland regions.

## 2.3 New dynamic surface cover scheme in LiBry

LiBry calculates the dynamic water content of a physiological strategy from the balance of water uptake by rainfall, snow melt, or dew, and water loss by evaporation. Using the biocrust type-specific relations between water saturation and NO and HONO emissions, it is therefore possible to quantify the emissions for each strategy at each time step of the simulation. To obtain

emissions per biocrust type, however, it is necessary to weight the emissions of each physiological strategy which belongs to the respective biocrust type by their relative abundance at a given location. In a second step, we therefore introduced a new scheme into LiBry which determines the relative abundances of the physiological strategies. A similar scheme was already developed for an earlier version of the model (Porada et al., 2016b). However, the strategies did not interact in the old scheme, they were all simulated independently for each location of the global grid, and subsequently their properties were weighted

by their productivity at the given location. While such an approach is appropriate for calculating large-scale average carbon fluxes per area, it is not ideal for computing the surface cover of the strategies. Without interaction, the surface cover of a strategy only depends on climatic conditions and disturbance events, which may not be realistic enough. In the new scheme, all strategies growing at the same grid point of the model have to share the available area for growth, and the weights for computing large-scale fluxes are thus their relative cover values (Fig. 3). Interactions between strategies then take place during

the expansion of the surface cover. New surface cover of a strategy depends on the growth of new area per area of the organism. To express this new area per area of the model grid, it is multiplied by the fraction of surface covered by the strategy at a given location of the grid. Expansion of surface cover is only possible into free area, which is not already covered by strategies. Thereby, we assume that the accessibility of the free area decreases as the fraction of free area becomes smaller at a given location. Hence, the strategies cannot contribute with their full surface cover to the formation and expansion of new cover,

but only with a part of their cover (Fig. 3 (a)). To account for direct competition between strategies, we redistribute the area which contributes to new cover among all strategies. Thereby, the weight of a strategy used for the redistribution corresponds to its height divided by the sum of the heights of all strategies. The model has reached a steady state with regard to surface

cover, when the expansion of cover of all strategies is compensated on average by the loss of cover due to disturbance (Fig. 3 (b)). The steady state spatial pattern of the total surface cover thus depends on spatial differences in growth due to climate, and also spatially differing disturbance frequencies. Without disturbance, the strategies would gradually cover the whole area of the model grid. The relative cover of a strategy, which is the share of a strategy on the total cover, depends on growth rate

5 per area and height of the strategy. The height is set as a constant property of each strategy, and the average growth rate will be constant in steady state. Thus, also the relative cover of each strategy, and, consequently, the relative cover of each biocrust type, will be constant in the steady state of a model simulation.

In the new surface cover scheme, the properties of the dominant simulated strategy, and, consequently, the predicted most abundant biocrust type, depend on environmental conditions in the following way: Under unfavourable conditions, such as low

10 water availability or frequent disturbances, total biocrust cover will be low and the success of the individual strategies will be mainly limited by their potential expansion rate. This favours small strategies such as cyanobacteria, since they need less growth of biomass per area for a given expansion rate than taller strategies. Under optimal conditions, in turn, total biocrust cover will be high and expansion will be limited by the available area. This favours tall strategies, such as mosses and lichens, since they have a higher share on the new cover than small strategies for the same given expansion rate (Grime et al., 1990).

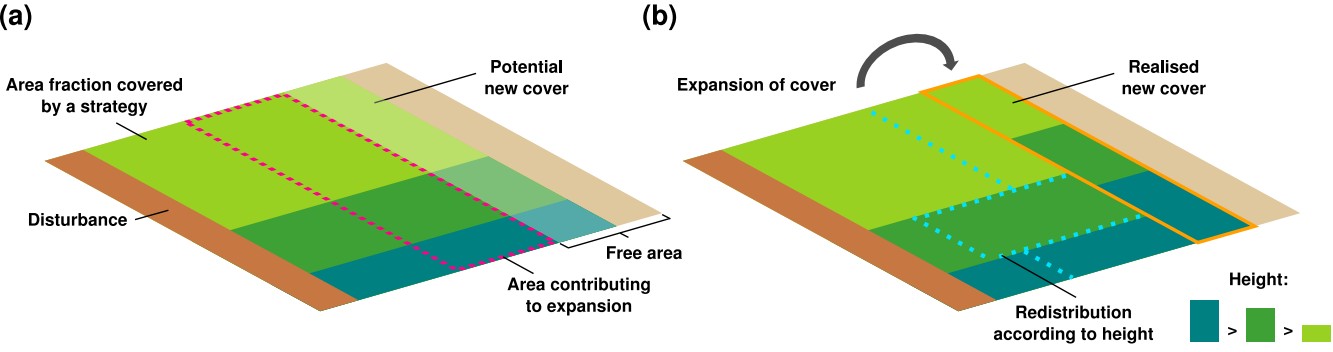

**Figure 3.** Dynamic surface cover scheme of the LiBry model. The surface cover of a strategy expands due to growth of biomass. The new cover is proportional to the area fraction covered by the strategy, and also to the growth rate of the strategy. For simplicity, the 3 strategies shown have the same growth rate. (a) only a part of the cover of a strategy can contribute to the new cover, indicated by the dotted magenta line, the remainder is not able to access the free area for growth. This contributing area decreases proportionally to the available free area. (b) to account for competition between strategies, the contributing area is redistributed according to the height of the strategies. Larger strategies have a higher weight in the redistribution than smaller ones, which represents their competitive advantage. Disturbance reduces the cover of all strategies by a certain fraction at a given time interval, leading to a steady state of cover where expansion is balanced by disturbance.

15 ## 2.4   Simulation setup

We ran the LiBry model for 600 years to achieve a steady state regarding biomass and surface cover, using an initial number of 3000 physiological strategies at each point of the model grid. Climate forcing data were based on the WATCH data set (see

Weedon et al. (2011) and also http://www.eu-watch.org/data_availability), which span the years 1958 to 2001. These years were repeated for the total length of the simulation (600 years). The WATCH data set comprises short-wave solar radiation, downwelling long-wave radiation, air temperature at 2 m height, rainfall, snowfall, wind speed at 10 m height, surface pressure and specific humidity. The latter two variables were used to determine relative humidity. The data have a temporal resolution of 3 hours, and they were interpolated to hourly resolution to match the hourly time step of the model, except for rainfall. To achieve a more realistic temporal distribution of rainfall, we used a random generator, which disaggregates daily sums of rainfall into hourly values, as described in Porada et al. (2016b). Maps of monthly leaf and stem area index and bare soil area were based on the Community Land Model (Bonan et al., 2002). They were used to compute the available area for growth of the organisms, and also the partitioning of radiation between the canopy and the ground. All data were spatially remapped to the resolution of 2.8125 times 2.8125 degrees (T42) of the rectangular model grid. In LiBry, intervals of different disturbance events are assigned to each point of the model grid based on the biome type at the respective location (see Tab. B4 and B5 in Porada et al. (2013)). This means, that for each of the 16 biomes which are considered in LiBry at the global scale, a characteristic interval between disturbance events was determined from the literature. Thereby, LiBry accounts for processes such as fire, windbreak of trees, which destroys the habitat of epiphytic non-vascular vegetation, and also trampling by animals. The disturbance interval is then converted into the fraction of the biocrust surface cover which is destroyed once per month in the simulation, e.g an interval of 8 years would result in roughly 1 % of the surface cover being disturbed each month. The biome classification is based on Olson et al. (2001). We constrained our simulation to regions belonging to the biomes desert, steppe, savanna, and mediterranean woodlands (Olson et al., 2001), where biocrusts are usually abundant. Furthermore, we excluded regions which exhibit more than 700 mm of annual rainfall, since biocrusts are usually outcompeted there by grasses and trees. To calculate global estimates, we averaged the last 20 years of the simulation for each point of the model grid.

## 2.5 Model validation and sensitivity analysis

We validated our modelling approach at two spatial scales: First, we compared the total biocrust cover simulated by LiBry for each location of the model grid to observed values of biocrust cover around the world, which were derived from the literature (Tab. A1). While our dataset of observed biocrust cover only includes a subset of the available studies, it is sufficient to obtain characteristic median values of cover for large regions. We used these median values, since the spatial coverage of field observations was not sufficient to directly determine the large-scale biocrust cover for each cell of the model grid. To estimate biocrust cover at large scale, empirical models have been used which, however, rely on correlations between cover and climatic conditions or soil properties to extrapolate field observations (Rodriguez-Caballero et al., 2018). To exclude these factors, we compared the model estimates to median values, although this means a spatially less detailed validation. In addition to the total cover, we validated the relative cover simulated by LiBry for the four biocrust types described above, also by comparison to median values of observations (Tab. A2).

Observations of average water saturation of biocrusts for long periods of time, and also field measurements of NO and HONO emissions, are quite rare. Hence, it was not possible to compute meaningful median values for these properties at the large scale. We therefore compared our simulated estimates to the available observations on an order-of-magnitude basis.

Secondly, we validated the LiBry model at the local scale. For this purpose, we used observations of total biocrust cover and relative cover of the four biocrust types from four study sites near Soebatsfontein, South Africa (Weber et al., 2015). We forced the LiBry model by climatic variables measured directly at the study sites, and we subsequently compared simulated cover to observations. Moreover, we evaluated our global climate data set, by selecting the grid point which includes Soebatsfontein, and

comparing simulated cover resulting from the global climate data to the cover resulting from the site climate data. To assess the water balance computed by LiBry, we compared simulated water content to observations of water content of biocrusts at Soebatsfontein. Since the maximum water storage capacity of the biocrusts was not determined in the field, we could only compare absolute water content, and not water saturation. These two quantities, however, are proportional to each other. Furthermore, we compared the surface temperature of the simulated biocrusts to field measurements in Soebatsfontein. We

also assessed the model performance for another local site, located close to Almería in South Spain (Raggio et al., 2017), where metabolic activity of biocrusts was observed, which is closely connected to their dynamic water content. Consequently, we compared simulated active time to the field observations. It should be mentioned that not all input variables needed for running the LiBry model were available from the two study sites. While the most important variables solar radiation, rainfall, air temperature, and relative humidity were available, we used atmospheric downwelling longwave radiation from our global

data set and, for Soebatsfontein, also wind speed. Moreover, we used leaf and stem area index from the global data, which are needed to compute shading effects, but are very low for the regions of Soebatsfontein and Almería and thus hardly affect the results. Due to lack of knowledge on the disturbance regime, we assumed the standard disturbance interval of 100 years used in LiBry for the desert biome. All data from Soebatsfontein were obtained between October 2008 and October 2009, and the data from Almería are from 2013. The generally low availability of data sets which include biocrust cover together with time

series of soil moisture, temperature, and climate variables limits our validation to two locations.

To assess the effects of uncertain parameter values on the model estimates, we performed a sensitivity analysis. The original model was already tested in this regard, and it was found that simulated NPP was not very sensitive to changes in various model parameters. This means that varying parameter values by $\pm50\%$ resulted in substantially smaller than 50% variation in estimated NPP in most cases (see Tab. 2 in Porada et al. (2013)). For this reason, we only analysed a subset of model parameters

here. We selected parameters which are supposed to have a significant impact on the main estimates of this study, namely total biocrust cover, relative cover of the four different biocrust types, and NO and HONO emissions.

(a) The simulated total surface coverage was previously tested to be most sensitive to the disturbance interval. Therefore, the disturbance frequencies for the biomes considered here were varied from the doubled value to half the value.

(b) The values of photosynthetic capacity and $CO_2$-diffusivity, which were used to distinguish between biocrust types,

directly affect the relative cover of the crust types. Hence, these values were increased and decreased by $20\%$, respectively.

(c) The $Q_{10}$-value, which controls the temperature dependence of NO and HONO-emissions by biocrusts, may markedly affect the simulated global emissions. It was therefore increased and decreased by a value of 0.5, respectively.

Additionally, we tested two uncertain parameters which may influence the dynamic water saturation of the simulated organisms, since this will also affect estimated NO and HONO emissions.

(d) The maximum amount of dew which can be collected by non-vascular vegetation in the model was varied from the doubled value to half the value. Limitation of dew formation in LiBry is necessary since the model does not simulate explicitly the dynamic water content of air in the atmosphere. The default value of $40\,\mathrm{mm\,yr^{-1}}$ in LiBry is based on observations of annual dew in drylands (e.g. Vuollekoski et al. (2015)).

(e) The resistance of the vegetation surface to evaporation of water was increased from 0 to $100\,\mathrm{s\,m^{-1}}$. Since non-vascular organisms have no active means to control water loss, such as stomata, no resistance against evaporation is assumed by default in LiBry. However, we cannot exclude the possibility that certain morphological features reduce evaporation, and thus we tested a resistance value which roughly corresponds to an average stomata conductance (Monteith, 1981).

Moreover, we accounted for uncertainty resulting from variation in the measured relations between water saturation and NO

and HONO emissions of different biocrust types.

(f) We calculated the standard deviation of the measurements made by (Weber et al., 2015), and subtracted it from the average curves shown in Fig. 1 to create a lower bound of NO and HONO emissions as a function of biocrust water content. To create a corresponding upper bound, we added one standard deviation to the curves.

(g) We replaced our default relationship between water content and NO and HONO emissions of different biocrust types by

an alternative one etablished by Meusel et al. (2018). They used a similar approach, but for a different location, a field site in Cyprus.

## 3   Results

### 3.1   Global patterns of biocrust cover and NO and HONO emissions

Emissions of NO and HONO by biocrusts have been shown to strongly depend on the type of biocrust (see Fig. 1). To take this

into account, we introduced a new scheme into the LiBry model which allows for a representation of different biocrust types and their associated NO and HONO emissions. Total NO and HONO emissions were then estimated by weighting the biocrust types by their relative abundances.

Spatial patterns of relative cover of the four types light cyanobacteria-, dark cyanobacteria-, chlorolichen-, and moss-dominated biocrust simulated by the LiBry model at the global scale are shown in Fig. 4 (a) to (d). Light cyanobacteria-

dominated biocrusts are abundant throughout all of the considered biomes desert, savanna, steppe, and mediterranean wood-lands, except for the driest parts of the world's large deserts, such as the Sahara or the Arabian desert, for instance. Interestingly, the relative cover of light cyanobacteria-dominated biocrusts increases with increasing dryness of regions up to a certain point, beyond which no simulated strategies are able to survive in the model. Dark cyanobacteria-dominated biocrusts show a spatial pattern which is similar to light ones, but their relative cover in areas close to extremely dry regions is smaller compared to light

cyanobacteria-dominated biocrusts. Chlorolichen-dominated biocrusts have a more constrained global distribution than light and dark cyanobacteria-dominated biocrusts in the model. They are, for instance, excluded from the dry inner part of Australia, and they also occupy smaller areas compared to light and dark cyanobacteria-dominated biocrusts in the Sahara and the Arabian desert. The spatial pattern of moss-dominated biocrusts is similar to chlorolichen-dominated biocrusts. Moss-dominated

biorusts are slightly less abundant than chlorolichen-dominated ones, except for a few regions in inner Australia. In general, moss- and chlorolichen-dominated biocrusts are more abundant in more poleward regions and light and dark cyanobacteria-dominated biocrusts are more abundant in regions at low latitudes.

The global pattern of the total cover of biocrusts in the biomes considered here is shown in Fig. 4 (e). According to our simulation, biocrusts cover 11 % of the global land surface. Interestingly, biocrust cover seems to be highest for desert regions, compared to other biomes, although biocrust cover tends to increase with rainfall within a biome. In Fig. 5 we show the dependence of the simulated relative cover of the different biocrust types on the amount of rainfall and the average temperature. In general, the relative cover fractions of light and dark cyanobacteria-dominated biocrusts increase with warmer and drier climatic conditions, while the share of chlorolichen- and moss-dominated biocrusts on the total coverage increases for cooler and wetter conditions. For regions with the lowest rainfall, only light cyanobacteria-dominated biocrusts occur in the model. If rainfall slightly increases, the relative cover of dark cyanobacteria-dominated biocrusts rises, until they are equally abundant as the light cyanobacteria-dominated crusts. The slope of this increase, however, depends on temperature (Fig. 5 (a,b)): Dark cyanobacteria-dominated biocrusts increase in abundance faster under cooler climatic conditions. The response of the relative cover fractions of chlorolichen- and moss-dominated biocrusts to temperature and rainfall is similar: They are both absent under the warmest and driest climatic conditions, but under cool and wet conditions they have the largest share on the total cover. Chlorolichen-dominated crusts seem to grow slightly better under cool and dry conditions compared to moss-dominated crusts (Fig. 5 (a,c)). The dependence of biocrust type on amount of rainfall and average temperature is also reflected in the average relative cover values per biome simulated by LiBry (see Tab. 1). In deserts, light cyanobacteria-dominated biocrusts show the highest relative cover, while in savanna and steppe regions, the four biocrust types are equally abundant, and woodlands show higher relative cover of chlorolichen- and moss-dominated biocrusts. Potential explanations for the simulated patterns of relative cover of the four biocrust types, the global pattern of total biocrust cover, and the correlation of cover types with rainfall are discussed below.

NO and HONO emissions do not only depend on the type of biocrust, but they are also strong nonlinear functions of the water saturation of the crust (see Fig. 1). Therefore, it is important that the model captures the temporal patterns of water saturation in a realistic way. Since time series of the water saturation of biocrusts are rarely determined in the field, the fraction of metabolically active time may be used instead as a measure of the hydrological dynamics of the crust. In Fig. 6 we show the simulated percentage of time which is spent in a metabolically active state, averaged over all physiological strategies at each location. The large-scale spatial pattern of active time shows highest values in more poleward regions, medium values in large parts of (sub)tropical regions and lowest values in desert regions. This may be explained by a combination of the patterns of rainfall and surface temperature (see Fig. A1): In poleward regions, rainfall is relatively high and evaporation is moderate, due to lower surface temperatures, resulting in relatively large water supply, as long as the water is not frozen. In many tropical regions, rainfall is also high but evaporation is markedly increased compared to more poleward regions, as indicated by the higher surface temperatures. This results in less available water and thus in less active time.

Based on simulated relative surface cover of biocrust types and their dynamical water saturation, we estimate global patterns of NO and HONO emissions by biocrusts, which are shown in Fig. 7. Highest values of NO and HONO emissions occur in

East Africa, around the Sahara, in North Australia, and in the hot deserts of North and South America. Northern, cooler regions show less strong NO and HONO emissions, although the percentage of active time is relatively high there. These patterns may be explained by the combined effects of ambient temperature and water saturation on NO and HONO emissions, which lead to high emission rates in warm conditions at intermediate to low water content. We estimate annual values of 1.04 Tg yr$^{-1}$ NO-N and 0.69 Tg yr$^{-1}$ HONO-N released by biocrusts at the global scale.

An overview of the results in form of global total values for different biomes and the whole study region is provided in Tab. 1.

| Region | Cover | Relative cover [%] | | | | Active | NO-N emissions | HONO-N emissions |
|--------|-------|-----|-----|-----|-----|--------|----------------|------------------|
|        | [%]   | LC  | DC  | CC  | MC  | time [%] | [Tg yr$^{-1}$] | [Tg yr$^{-1}$] |
| Global | 30 | 39 | 23 | 21 | 17 | 19 | 1.04 | 0.69 |
| Desert | 28 | 50 | 20 | 18 | 12 | 14 | 0.47 | 0.31 |
| Grassland | 32 | 27 | 24 | 27 | 22 | 24 | 0.49 | 0.33 |
| Woodlands | 35 | 23 | 13 | 34 | 30 | 28 | 0.083 | 0.053 |

**Table 1.** Annual global total values of biological soil crust cover, relative cover of biocrust types, metabolically active time, and NO and HONO emissions estimated by LiBry. 'Grassland' means steppe and savanna biomes, 'Woodlands' means mediterranean woodland biomes. The abbreviations in the second row stand for (LC) light cyanobacteria-, (DC) dark cyanobacteria-, (CC) chlorolichen-, (MC) moss-dominated biocrusts. Units are shown in rectangular brackets. Note that NO and HONO emissions are global sums, while the other properties are average values for the respective regions. For relative cover and active time, regions where no strategies were able to survive in the model were excluded. Total biocrust cover in deserts is relatively low due to large areas where no simulated strategies survive. If these areas were excluded, biocrust cover in deserts would amount to 45 %.

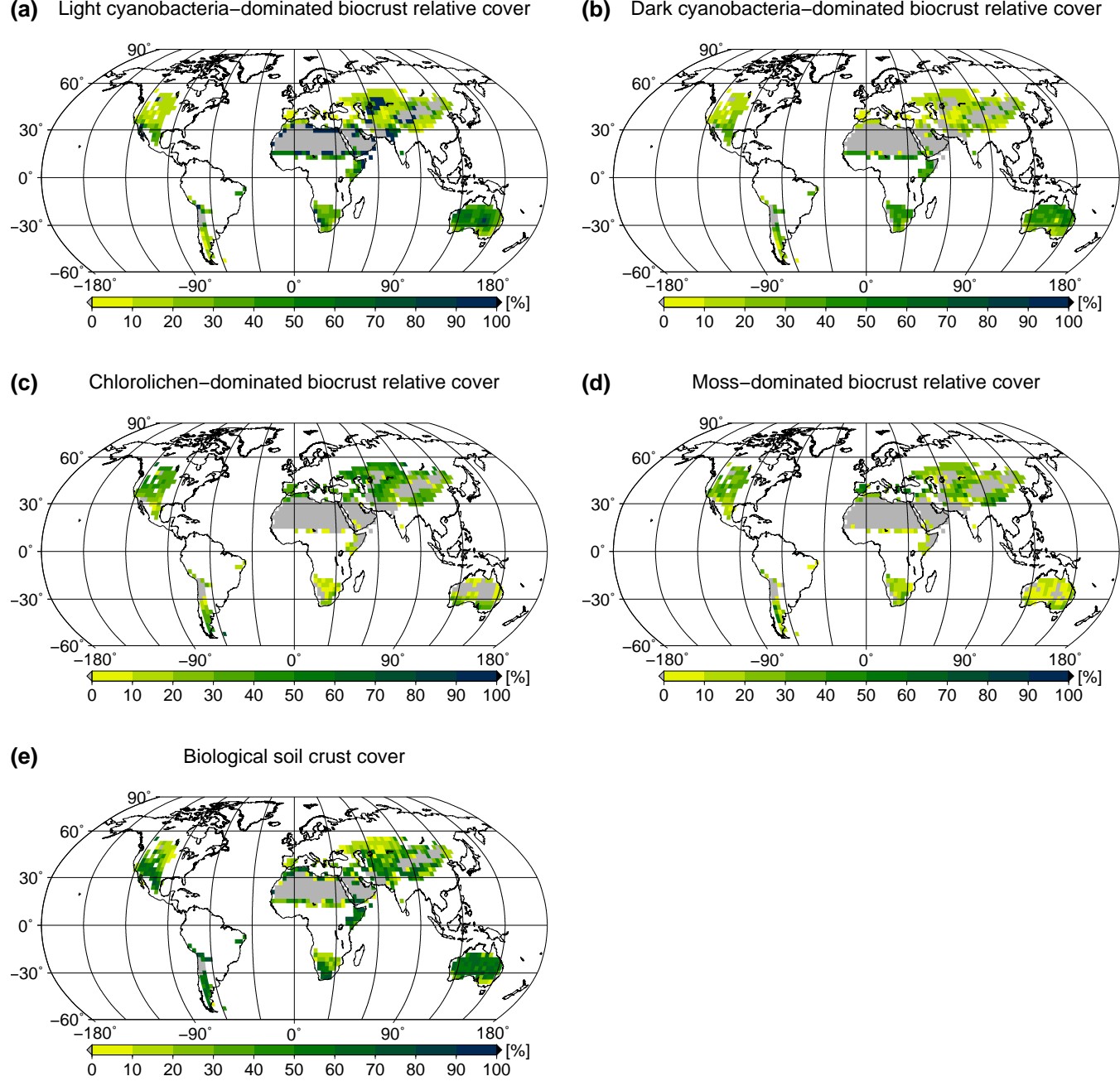

**Figure 4.** Global patterns of biological soil crust cover simulated by the LiBry model. Relative cover of (a) light cyanobacteria- (b) dark cyanobacteria- (c) chlorolichen- and (d) moss-dominated biocrusts is shown, and also (e) total biocrust surface cover. White areas at the land surface denote regions which are excluded since they do not belong to the biomes desert, steppe, savanna, and mediterranean woodlands, or exhibit more than 700 mm of annual rainfall. Grey areas denote regions where no simulated strategies are able to survive, although they match the above criteria for biocrust cover.

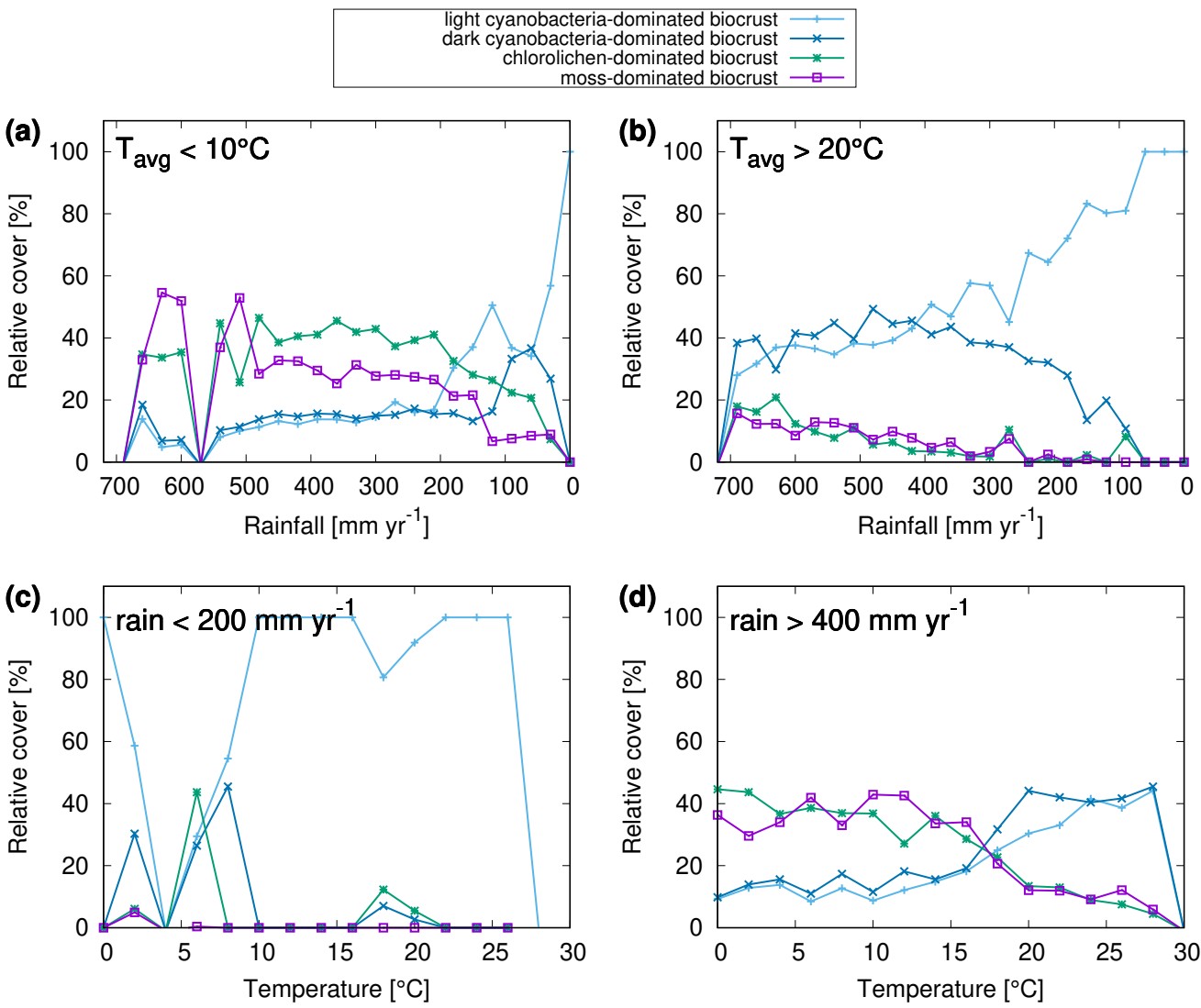

**Figure 5.** Dependence of relative cover of different biocrust types on annual rainfall and average temperature. (a - d) Only a subset of all grid cells in the study area are included in the relations, depending on the climatic conditions specified in the upper left corner of the plots. The point symbols (boxes, crosses) denote average values of relative cover for bins of rainfall with a range of 30 mm yr$^{-1}$. Regions with zero biocrust cover are excluded from the data shown in (a - d).

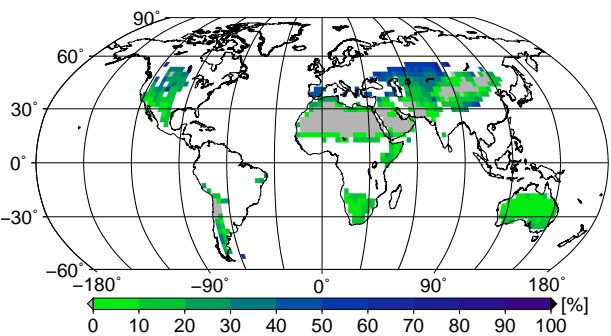

**Figure 6.** Global pattern of the percentage of time spent in a metabolically active state per year, simulated by LiBry. White areas denote excluded regions, while grey areas correspond to regions where no simulated strategies are able to survive.

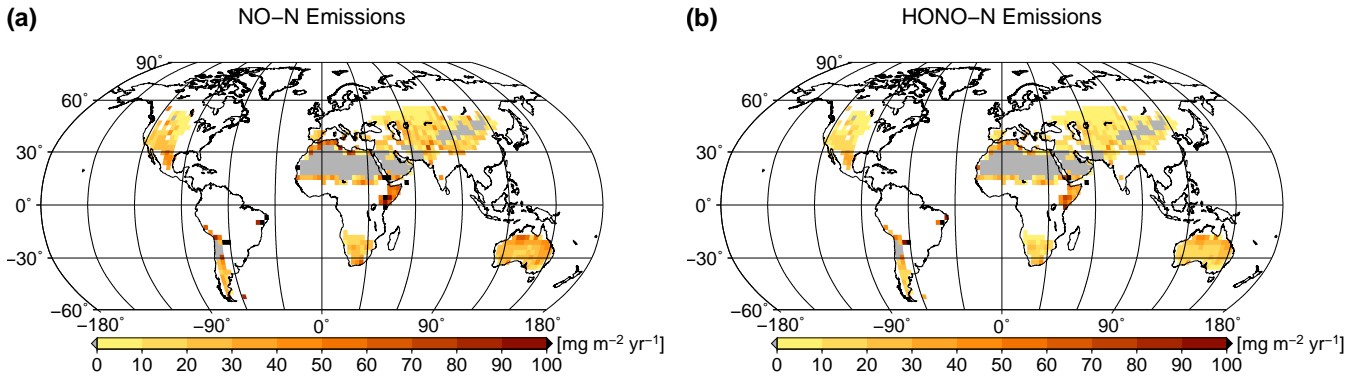

**Figure 7.** Global patterns of (a) NO and (b) HONO emissions simulated by LiBry, in units of $mg\,m^{-2}\,yr^{-1}$ of nitrogen. White areas denote excluded regions, while grey areas correspond to regions where no simulated strategies are able to survive.

## 3.2 Global validation

We validate our modelling approach both at the global scale and the local scale. In Fig. 8 (a) simulated biocrust cover for the biomes desert, savanna/steppe, and mediterranean woodlands, and also for the whole considered area are shown together with observed biocrust cover from the literature. Simulated cover matches well to median values of observations, and also the large range of the observations is reproduced well by the model. The higher biocrust cover in deserts compared to savanna and steppe regions, which is predicted by the LiBry model (see also Fig. 4 (e)) is also reflected in the median values of the observations. Furthermore, our simulated value of total biocrust cover of 11 % of the global land surface area is in good agreement with the value of 12 % estimated by a recent empirical large-scale study on the global extent of biocrusts (Rodriguez-Caballero et al., 2018).

Figure 8 (b) shows simulated average relative cover of light cyanobacteria-, dark cyanobacteria-, chlorolichen- and moss-dominated biocrusts at the global scale together with observations from the literature. The model reproduces well the sequence of the median values of relative cover: light cyanobacteria- >dark cyanobacteria- and chlorolichen- >moss-dominated biocrusts. The large range of observations is also represented by the model. Compared to the median values, LiBry underestimates relative cover of light cyanobacteria-dominated biocrusts, while relative cover values of dark cyanobacteria-, chlorolichen-, and moss-dominated biocrusts are slightly overestimated.

Regarding active time and NO and HONO emissions, only a few studies report observational data from field experiments. Büdel et al. (2018) estimate a value of 24 % of metabolically active time of biocrusts throughout the year in a savanna ecosystem in North Australia. The LiBry model predicts between 10 and 20 % active time for the region which includes this ecosystem, which is consistent with the field measurements. Meusel et al. (2018) estimate average combined NO-N and HONO-N emissions of 160 mg m$^{-2}$ yr$^{-1}$ originating from the land surface in Cyprus, of which 28-46 % can be attributed to biocrusts, depending on climatic conditions. Cyprus is not represented by the LiBry model due to the relatively coarse resolution of the model grid. However, for the close-by area of South Turkey the model predicts combined NO and HONO emissions of 30-50 mg m$^{-2}$ yr$^{-1}$, which agrees well with the empirical estimate by Meusel et al. (2018). Barger et al. (2005) estimate annual NO emissions of 2 to 16 mg m$^{-2}$ yr$^{-1}$ by biocrusts at two fields sites in the Canyonlands National Park, Utah, USA. This compares well to the large-scale estimate of 10 to 20 mg m$^{-2}$ yr$^{-1}$ of NO-N simulated by LiBry for this region.

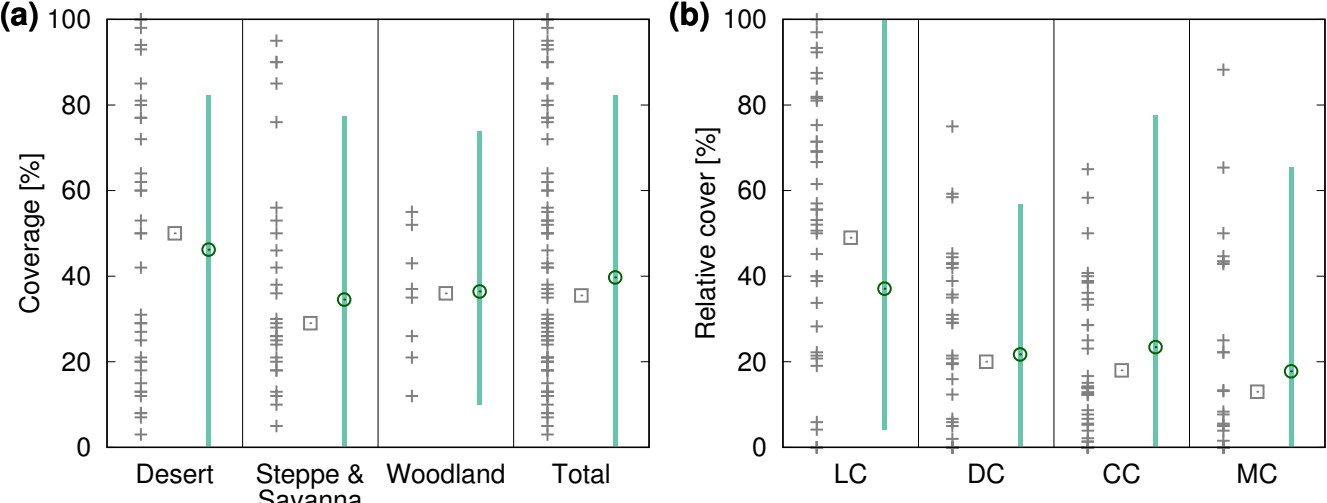

**Figure 8.** Comparison of biocrust cover estimated by LiBry to field observations (a) for 3 different biomes and the total study region and (b) for 4 different biocrust types. The abbreviations on the x-axis stand for (LC) light cyanobacteria-, (DC) dark cyanobacteria-, (CC) chlorolichen-, (MC) moss-dominated biocrusts. The grey crosses show field measurements of biocrust cover from various studies, which are listed in the supplement in Tab. A1 and A2. Grey rectangles correspond to the median values of all field measurements in a column. The green circles show (a) average simulated biocrust cover for all points of the model grid belonging to a biome and (b) average relative cover for all points of the study region, separated into biocrust types. Green bars denote the range of (a) cover and (b) relative cover of all considered grid points. Note that regions where no strategies were able to survive in the model are excluded from the comparison.

### 3.3 Local validation

In Fig. 9 we compare simulated cover to field observations from four study sites near Soebatsfontein, South Africa (Weber et al., 2015). The LiBry model, which is forced by climate data measured near Soebatsfontein, reproduces well the total biocrust cover at the sites, and also the sequence of relative cover values for the four biocrust types. Relative cover of dark cyanobacteria-
5   dominated biocrusts, however, is underestimated by the model, while relative cover of chlorolichen-dominated biocrusts is overestimated. The field observations show a larger spread between the four sites than the model estimates. Additionally, we compare the observations to relative cover based on a simulation forced by global climate data from the region which includes Soebatsfontein (Fig. 9). These LiBry estimates reproduce observed values of cover as well as the LiBry estimates which are based on the locally measured climate data.
10    Figure 10 (a) shows observed water content of biocrusts together with values simulated by LiBry for Soebatsfontein. The model captures well the timing and magnitude of moisture content, while the durations of the moist periods are slightly underestimated for larger rainfall events. In Fig. 10 (b) we compare surface temperature of biocrusts simulated by LiBry for the Soebatsfontein area to field measurements. The model reproduces well the annual cycle of surface temperature and

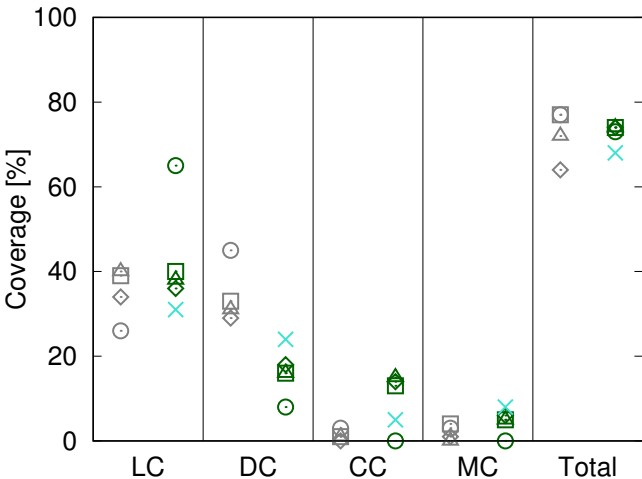

**Figure 9.** Comparison of biocrust cover estimated by LiBry to observations at four different field sites at Soebatsfontein, South Africa. Absolute values of cover of the four different biocrust types and also total cover are shown. The abbreviations on the x-axis stand for (LC) light cyanobacteria-, (DC) dark cyanobacteria-, (CC) chlorolichen-, (MC) moss-dominated biocrusts. The field sites are distinguished by different geometric symbols. Grey symbols on the left side of each column denote field observations, while green symbols on the right correspond to model estimates based on climate data measured at the field sites. Blue crosses show model estimates obtained from global climate data for the Soebatsfontein region.

the magnitude of daily variations in temperature. Simulated surface temperature in the warm season is, however, slightly underestimated. Figure 10 only shows one biocrust type for one of the four field sites in Soebatsfontein for clarity. The complete overview of simulated dynamic water content and surface temperature compared to field observations can be found in the supplement (Fig. A2 to A5).

5    Figure 10 (c,d) shows observed metabolically active time of biocrusts together with estimates simulated by LiBry for Almería. The observed monthly pattern of active time is well reproduced by the model (Fig. 10 c), and also for the daily pattern, the model agrees with the measurements (Fig. 10 d). While the response to rain events is captured in general, the model predicts several small to moderate peaks in activity, which do not occur in the observations. Moreover, the model does not entirely reproduce observed periods of prolonged activity in late spring and winter, which leads to a slight underestimation
10  of total annual activity by the model. These findings are discussed below.

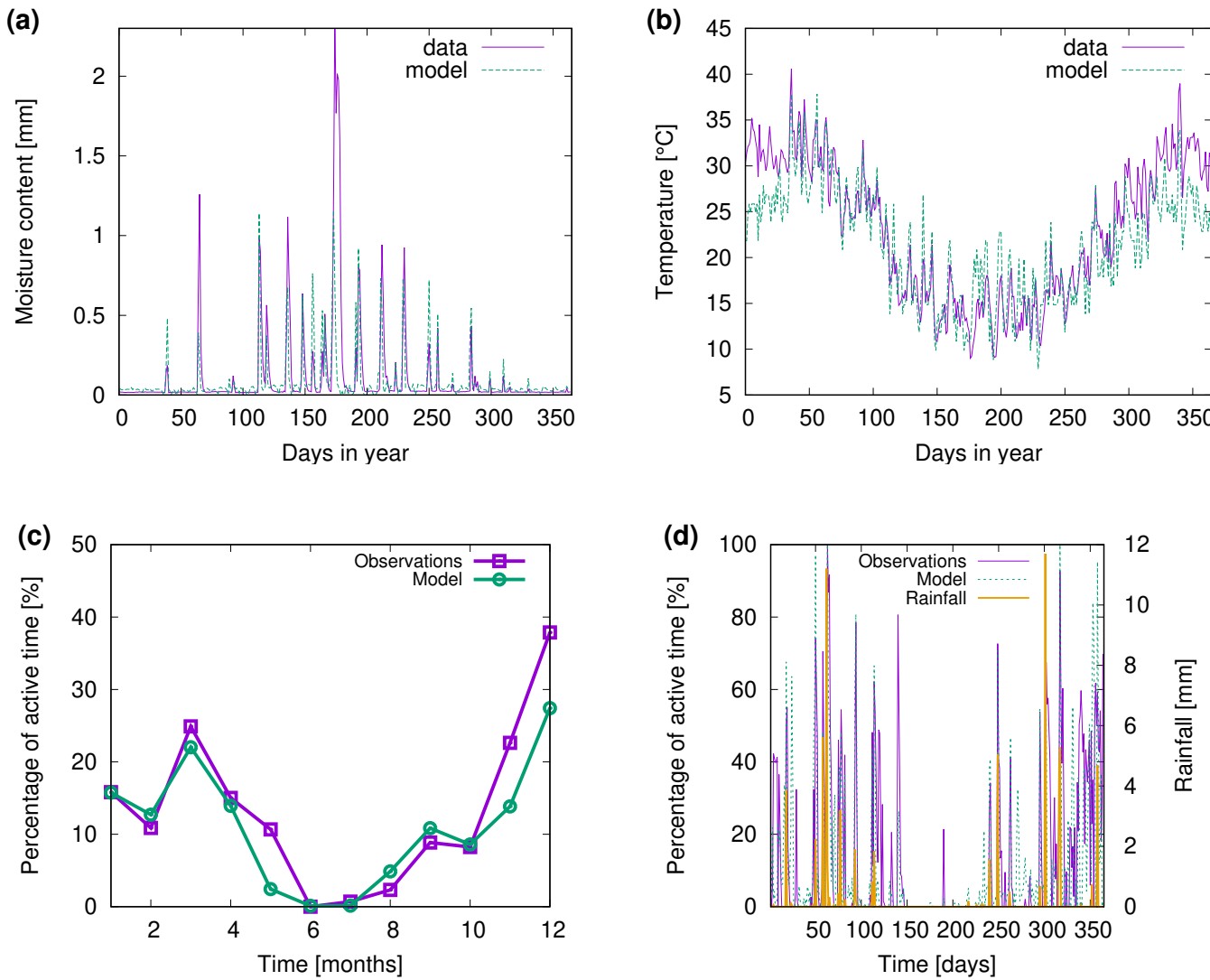

**Figure 10.** Comparison of (a) biocrust water content and (b) surface temperature estimated by LiBry to observations for one field site (number 4) and one biocrust type (dark cyanobacteria-dominated biocrust) at Soebatsfontein, South Africa. Comparison of (c) monthly and (d) daily biocrust active time fraction simulated by LiBry to observations from a field site near Almería, South Spain.

## 3.4 Sensitivity analysis

In Tab. 2 we show the outcome of our sensitivity analysis, where we tested how the variation of uncertain parameter values affects the global estimates. We selected parameters which likely affect total biocrust cover, the relative cover of the four different biocrust types, and NO and HONO emissions. Total cover slightly decreases for shorter disturbance intervals and increases for longer intervals, which is expected. Shifting the values of $CO_2$-diffusivity and photosynthetic capacity, which determine the biocrust type leads to changes in the relative cover of the respective biocrust types in the expected direction. Changes in the $Q_{10}$-value of NO and HONO emissions have no effect on biocrust cover, since the emissions do not feed back on the organisms in the model. However, NO and HONO emissions seem to be inversely related to their $Q_{10}$-value, which is discussed further below.

Biocrust cover is relatively sensitive to changes in the amount of available dew. While a reduction in dew mostly affects the relative cover fractions of the different crust types, a doubling of dew increases total simulated cover from 30% to 41%, active time slightly increases and estimated NO and HONO emissions more than double. Furthermore, active time and, consequently NO and HONO emissions by biocrusts are sensitive to increasing the resistance to evaporation. Active time increases by 10%, and NO and HONO emissions by 48%.

In general, our global results show low sensitivity to variation of the selected parameters, which means that the estimates change substantially less than the varied parameters on a relative basis. Reducing the disturbance interval by half, for instance, only leads to a 10% reduction in total biocrust cover, and doubling it causes only a 7% increase. Compared to this, shifting the threshold values for the assignment of physiological strategies to certain biocrust types has a larger effect on the relative cover fractions of the four biocrust types. However, we find a relatively large sensitivity of estimated NO and HONO emissions to parameters which control active time and the relationship between water saturation and emissions. This is discussed below in more detail.

Simulated NO and HONO emissions are also sensitive to variation in the relationship between water content and emissions. Decreasing the specific emissions at a given water content by one standard deviation reduces simulated total global NO emissions by 25% and HONO emissions by 50% compared to the originally estimated emissions (see Tab. 2). Increasing specific emissions by one standard deviation raises simulated NO emissions by 85% and HONO emissions by 107%. Moreover, replacing our default relation between water content and NO and HONO emissions, which is based on Weber et al. (2015), by an alternative relation derived from Meusel et al. (2018), significantly affects our estimates. Simulated NO emissions decrease by 73%, while HONO emissions are only reduced by 17%.

## 4 Discussion

In this study we estimated NO and HONO emissions by biological soil crusts (biocrusts) at the global scale using the process-based non-vascular vegetation model LiBry. Thereby, emission rates of NO and HONO were based on water saturation of four distinct biocrust types. LiBry quantified spatial and temporal patterns of relative cover and saturation for each biocrust type and thus computed large-scale emissions.

| Simulation | Cover [%] | Relative cover [%] | | | | Active time [%] | NO-N emissions [Tg yr$^{-1}$] | HONO-N emissions [Tg yr$^{-1}$] |
|---|---|---|---|---|---|---|---|---|
| | | LC | DC | CC | MC | | | |
| Control | 30 | 39 | 23 | 21 | 17 | 19 | 1.04 | 0.69 |
| $\tau_D$ x 0.5 | 27 | 36 | 28 | 20 | 16 | 19 | 0.98 | 0.66 |
| $\tau_D$ x 2.0 | 32 | 40 | 19 | 23 | 18 | 19 | 1.10 | 0.72 |
| $D_{CO2}$ - 20 % | 30 | 39 | 23 | 12 | 26 | 19 | 1.03 | 0.72 |
| $D_{CO2}$ + 20 % | 30 | 39 | 23 | 29 | 9 | 19 | 1.05 | 0.67 |
| $PSC_{CO2}$ - 20 % | 30 | 36 | 25 | 22 | 17 | 19 | 1.07 | 0.72 |
| $PSC_{CO2}$ + 20 % | 30 | 49 | 12 | 22 | 17 | 19 | 0.98 | 0.62 |
| $Q_{10}$ - 0.5 | 30 | 39 | 23 | 21 | 17 | 19 | 1.24 | 0.81 |
| $Q_{10}$ + 0.5 | 30 | 39 | 23 | 21 | 17 | 19 | 0.99 | 0.66 |
| $dew_{MAX}$ x 0.5 | 30 | 33 | 30 | 23 | 14 | 19 | 1.07 | 0.73 |
| $dew_{MAX}$ x 2.0 | 41 | 55 | 19 | 17 | 9 | 22 | 2.50 | 1.74 |
| $r_S 100$ | 33 | 38 | 23 | 23 | 16 | 23 | 1.54 | 1.02 |
| $E_{LB}$ | 30 | 39 | 23 | 21 | 17 | 19 | 0.78 | 0.34 |
| $E_{UB}$ | 30 | 39 | 23 | 21 | 17 | 19 | 1.92 | 1.43 |
| $E_{CYP}$ | 30 | 39 | 23 | 21 | 17 | 19 | 0.28 | 0.57 |

**Table 2.** Impact of varied parameter values on annual global total values of biocrust cover, relative cover of biocrust types, metabolically active time, and NO and HONO emissions estimated by LiBry. The abbreviations in the second row stand for (LC) light cyanobacteria-, (DC) dark cyanobacteria-, (CC) chlorolichen-, (MC) moss-dominated biocrusts. 'Control' means the control run, which is also shown in Tab. 1. '$\tau_D$' denotes the disturbance interval, which is multiplied by 0.5 and 2.0. '$D_{CO2}$' and '$PSC_{CO2}$' stand for the value of $CO_2$-diffusivity and photosynthetic capacity, respectively, which are used to distinguish between crust types. Both parameters are increased and decreased by 20 %. '$Q_{10}$' represents the temperature dependence of NO and HONO-emissions and it is increased and decreased by a value of 0.5. '$dew_{MAX}$' corresponds to the maximum amount of dew per year, which is multiplied by 0.5 and 2.0. '$r_S 100$' denotes the surface resistance of the vegetation against evaporate, which is increased from 0 to 100 s m$^{-1}$. '$E_{LB}$' and '$E_{UB}$' stand for the lower and upper boundaries, respectively, of the relation between water content and NO and HONO emissions. '$E_{CYP}$' corresponds to an alternative relation between water saturation and NO and HONO emissions of biocrust types.

## 4.1 Cover and biocrust types

We found a dependence of biocrust type on the amount of rainfall and average temperature (Fig. 5). In the driest regions, light cyanobacteria-dominated biocrusts are most abundant. With increasing rainfall, the relative cover of dark cyanobacteria-dominated crusts increases, followed by chlorolichen- and moss-dominated biocrusts. With decreasing temperature, chlorolichen- and moss-dominated biocrusts become increasingly abundant. At cool and wet conditions, they are more abundant than light and dark cyanobacteria-dominated biocrusts, which has also been reported from field studies (Bowker et al., 2006; Kidron et al., 2010; Büdel et al., 2009).

To understand why the LiBry model predicts these climate-driven spatial patterns of different physiological processes have to be considered, which are implemented in the model. First, the dynamic cover scheme requires all strategies to compensate losses in their surface cover through growth, in order to survive. These losses result from disturbance and turnover of biomass and they are not directly related to climate. Growth, however, is proportional to the difference between photosynthesis and respiration, which means that it decreases with drier conditions, which reduce the amount of active time, and it also decreases with higher temperatures, since respiration increases with temperature in the model. Consequently, markedly dry and warm conditions are unfavorable in general for the simulated strategies. Secondly, smaller strategies have the advantage of a more efficient cover expansion in the model, since they can produce more surface area for a given amount of biomass growth than taller strategies. This means that they are more likely to maintain their cover against disturbance and turnover under unfavorable climatic conditions. Since we defined small strategies (less than 2 mm) as cyanobacteria, this explains the high relative cover of light- and dark cyanobacteria-dominated biocrusts under warm and dry conditions. Fig. 5 shows that dark cyanobacteria-dominated crusts do not perform as well as light cyanobacteria-dominated crusts under the warmest and driest conditions. This can be explained by another physiological trade-off implemented in the model, which links photosynthetic capacity to maintenance respiration. We have defined that small strategies with a high photosynthetic capacity and thus a high respiration rate belong to dark cyanobacteria-dominated crusts, while those with low photosynthetic capacity and respiration are assigned to light cyanobacteria-dominated crusts (Fig. 2). Since respiration increases stronger than photosynthesis at high temperatures, strategies with a high 'baseline' respiration may grow less under warm conditions, which explains why dark cyanobacteria-dominated crusts are less abundant in the warmest regions. Moreover dark cyanobacteria-dominated crusts are more negatively affected by reduced active time due to dry climate compared to light cyanobacteria-dominated biocrusts, also at moderate temperatures (Fig. 2 (c)). A potential reason for this is the increase in turnover rate with higher photosynthetic capacity, which is not reduced as strongly as growth during periods with low activity. This means that dark cyanobacteria-dominated crusts can use their potential for stronger growth due to their higher photosynthetic capacity only if sufficient active time is available (and sufficient radiation). For this reason, their relative cover is maximal under moderate temperatures and sufficiently wet climatic conditions (Fig. 2 (d)). Finally, the growth height of a simulated strategy represents a competitive advantage in the model. If two strategies compete for the same location of available free area (Fig. 3), the taller strategy will be able to overgrow the smaller one. Hence, under favourable climatic conditions (moderate temperatures and sufficient rainfall), simulated lichens and bryophytes may partly outgrow cyanobacteria, and therefore chlorolichen- and moss-dominated biocrusts have the largest share on the total cover (Fig. 5). The higher relative cover of chlorolichen-dominated biocrusts compared to moss-dominated crusts at cooler and drier conditions (Fig. 5 (a,c)) may be explained by another physiological trade-off in the LiBry model: Strategies with a high diffusivity for $CO_2$ during the saturated state can grow more under favourable conditions than strategies with a low $CO_2$-diffusivity. However, due to their more open structure, they also evaporate more water, and thus become limited faster under dry conditions and are less productive. Since we have defined in the model that mosses have a higher $CO_2$-diffusivity and higher evaporation than lichens on average, the simulated moss-dominated biocrusts may be more affected by dry climate and are thus less abundant than chlorolichen-dominated crusts. This pattern is more pronounced for cool climatic conditions compared to warm conditions (Fig 5 (a,b)). A possible reason for this is that the higher resistance of

lichens against evaporation is not sufficient to prevent desiccation under warm climatic conditions, which means that they have no advantage over mosses anymore. Large water compensation and saturation values of bryophytes have also been experienced during $CO_2$ gas exchange measurements under controlled conditions (Tamm et al., 2018; Raggio et al., 2018).

Another finding is that simulated biocrust cover in desert regions is generally higher than in savanna and steppe regions, which is also supported by observational data (Fig. 8 (a)). This may be explained by different disturbance regimes in these biomes, which are also taken into account in the LiBry model. More frequent disturbances due to fire and grazing in steppe and savanna biomes seem to have a more negative impact on biocrust cover than the lower potential for growth in deserts. Furthermore, competition by grasses reduces the potential cover in steppe and savanna biomes, which is represented in LiBry by a decreased fraction of area available for growth in these regions. Negative effects of multiple types of disturbance and also the increasing competition of vascular plants at increasing annual precipitation rates have been described in multiple studies (Zaady et al., 2016; Zhang et al., 2016).

The comparison of simulated biocrust cover to field observations shows a good agreement in general, both at the global scale as well as the local scale of the Soebatsfontein field sites. At the global scale, the model underestimates relative cover of light cyanobacteria-dominated biocrusts, while the other biocrust types are slightly overestimated (Fig. 8 (b)). At the local scale, however, relative cover of light cyanobacteria-dominated biocrusts is well reproduced for 3 out of 4 field sites, while dark ones are underestimated and chlorolichen ones overestimated (Fig. 9). There are several explanations for these findings: First, the scheme we use to distinguish between different biocrust types may not reflect physiological properties of real biocrusts with high accuracy. However, we are not aware of studies which provide representative values for height, photosynthetic capacity and $CO_2$-diffusivity of the biocrust types considered here at the global scale. This uncertainty is taken into account by the simple scheme used here, which divides simulated ranges of photosynthetic capacity and $CO_2$-diffusivity in half. We think this method is more appropriate than calibrating the threshold values for these properties in a way that relative cover of biocrust types is accurately reproduced by the model. By selecting a slightly higher threshold value for photosynthetic capacity, for instance, we could have increased the relative cover of chlorolichen-dominated biocrusts in inner Australia compared to moss-dominated biocrusts. However, the relative cover of the different biocrust types simulated by LiBry does not only depend on the two physiological properties, $CO_2$-diffusivity and photosynthetic capacity, which were used to distinguish the crust types, but also on other factors, such as disturbance interval, for instance. By calibrating the model with regard to $CO_2$-diffusivity and photosynthetic capacity, we would implicitly assume that all other uncertain parameters have the correct values, which is not necessarily true.

Secondly, the number of observations of relative biocrust cover is relatively low and shows a large variation (Fig. 8 (b)). Hence, the median values of relative cover may not reflect real abundances of cover types very accurately. Finally, regarding the field sites at Soebatsfontein, the simulated disturbance interval or shading by vascular vegetation may not represent well the actual conditions at the sites, leading to biased cover estimates.

We find that the global climate data for the grid cell which includes Soebatsfontein are a good approximation to the climate data from the local station (see also Fig. A6). Consequently, the estimated values of cover of biocrust types based on the different climate input data are similar. However, for the site near Almería, the local data represent significantly drier conditions

than the large-scale data for South Spain, which explains the higher activity simulated by LiBry for Spain in general compared to the Almería site. This illustrates that the global data are not necessarily a good approximation for those local conditions which stronlgy deviate from the regional climate.

The sensitivity analysis shows that relative cover of light cyanobacteria-dominated biocrusts increases at the expense of dark ones for increasing disturbance intervals. This seems counterintuitive, since less frequent disturbances should lead to a higher relative cover of dark cyanobacteria-dominated biocrusts. However, this effect is overruled by the increase in total biocrust area under longer disturbance intervals, which is populated mainly by light cyanobacteria-dominated biocrusts, thus increasing their cover relative to dark ones.

## 4.2 Active time and surface temperature

Regarding active time of biocrusts, a global-scale evaluation of the model estimates is difficult due to lack of suitable field observations. At the field sites of Soebatsfontein, the pattern of simulated water content matches well the observations (Fig. 10). However, the durations of the moist periods are slightly underestimated, which means that also metabolically active time may be underestimated. This may be explained by the fact that the moisture sensors which were installed at the field sites measure the water content of the uppermost 5 mm of the whole biocrust, which also include fungi, soil bacteria and mineral soil. The LiBry model, however, only considers the water storage capacity of the photoautotrophic organisms, which is smaller than the whole biocrust storage capacity. Due to lower water storage capacity, the photoautotrophs may become desiccated earlier than the whole soil, leading to a shorter simulated active time compared to the observations at Soebatsfontein.

Simulated surface temperature of biocrusts is reproduced well by LiBry compared to observations at Soebatsfontein. For the warm season, however, surface temperature is underestimated by the model. This may result from the fact that protection of biocrusts by dark pigments against UV radiation is not considered in the model. Hence, simulated strategies with higher albedo than observed in the field may be selected in the model, which exhibit lower surface temperatures to reduce respiration losses. However, this is not of great importance for our main results, since the biocrusts are mostly inactive in the warm season. Thus, underestimated temperature values do not affect the simulated annual NO and HONO emissions to a large extent.

At the field site of Almería, the simulated monthly and daily patterns of active time match well the observations in general. However, in spring and fall, the model predicts several small and a few larger peaks of activity which cannot be seen in the measurements. One possible explanation for this is that the model may overestimate dew input from the atmosphere, possibly due to the spatially uniform value of maximum dew which is used in LiBry for all dryland regions. In contrast, the model underestimates longer periods of high activity in late spring and winter, which are not directly related to rainfall events (Fig. 10 d). Potential reasons for this may be the missing water storage capacity of the soil in the model, or underestimation by the model of activation from unsaturated air at relatively high humidity. Although the latter process is represented in the model, it contributes little to the total simulated water supply. Hence, same as for Soebatsfontein, LiBry tends to slightly underestimate active time.

The relatively large sensitivity of our estimated NO and HONO emissions to parameters which influence the dynamic water content is plausible. If maximum available dew is doubled, or a resistance to evaporation is introduced, the potential area where

biocrusts may occur in the model is significantly larger and total biocrust cover and active time significantly increase (Tab. 2). The strong increase in NO and HONO emissions can further be explained by the large increase in biocrust cover in warm regions, where high temperatures further enhance NO and HONO release (Eq. 1). However, the simulated global patterns of biocrust coverage under doubled dew or increased surface resistance seem to be inconsistent with observations. In extremely

dry regions of North Africa, Arabia and Australia, high cover values of more than 70% are simulated, which are substantially higher than large-scale average values which are commonly assumed for these regions (see Fig. A7). Moreover, although dew of 80 mm yr$^{-1}$ may occur locally under certain conditions in drylands, this value is most likely too high for a large-scale estimate (see also Vuollekoski et al. (2015) for typical values),

## 4.3 NO and HONO emissions

The LiBry model estimates global annual total values of 1.04 Tg yr$^{-1}$ NO-N and 0.69 Tg yr$^{-1}$ HONO-N released by biocrusts. These values are in good agreement with the 1.1 Tg yr$^{-1}$ NO-N and 0.6 Tg yr$^{-1}$ HONO-N estimated by Weber et al. (2015) using an empirical upscaling of field measurements. However, it should be noted that Weber et al. (2015) include a considerably larger area in their study, since they do not constrain potential biocrust extent by high rainfall. Hence, our approach predicts higher NO and HONO emissions per area than the empirical one (see Fig. A8 for an estimate without the rainfall constraint).

There are several factors that may lead to an underestimation of emissions by the empirical approach: First, emissions of NO and HONO were not computed directly from water saturation of biocrusts by Weber et al. (2015). Instead, they assumed that each rainfall event lead to a full wetting and drying cycle of the biocrust. By measuring the integrated NO and HONO emissions during one average wetting and drying cycle in the laboratory, and subsequently multiplying these emissions by the number of precipitation events, global emissions were estimated. However, comparison with field observations suggests that the number of

precipitation events derived from global rainfall data may have been considerably underestimated, due to the coarse 3-hourly temporal resolution of the dataset. Since we disaggregate the 3-hourly rainfall data to hourly data using a stochastic model (Porada et al., 2016b), we consider a higher number of rainfall events, which may partly explain our higher per-area estimates. Thereby, also small rainfall events, which do not cause a full wetting and drying cycle, may result in high NO and HONO emissions, due to the non-linear dependence of emissions on water saturation (Fig. 1). Secondly, globally uniform values of

relative coverage of the four different biocrust types were assumed in order to upscale NO and HONO emissions to global values. However, if the relative cover of the biocrust types correlates with global patterns of rainfall, which is suggested by LiBry results, the upscaling will be biased, since the biocrust types differ in their NO and HONO emission rates. It is difficult to estimate if this would result in an under- or overestimation of NO and HONO emissions: Dark cyanobacteria-dominated biocrusts, which were shown to be the most effective emitters of NO and HONO, tend to have higher coverage values at lower

annual precipitation amounts, but their peak emissions only require a water saturation of around 20 %. To summarise, our current approach allowed us to model biocrust emissions in a much more detailed manner, but, in the long run, additional field measurements would be helpful to further corroborate these results.

     Alternatively, our modelling approach may overestimate the true NO and HONO emissions at large scale. A potential reason for this may be overestimation of the simulated water saturation of dark cyanobacteria-dominated biocrusts during their periods

of metabolic activity. Since NO and HONO emissions of dark cyanobacteria-dominated biocrusts show a strong peak at a water saturation of approximately 20 %, emissions would be overestimated if this value was simulated too frequently by the model (Fig. 1). However, Fig. 10 (a) suggests that the LiBry model correctly predicts the number of wetting events. To assess this further, field measurements of water saturation would be needed, which allow to reconstruct the temporal distribution of the degree of saturation. Another reason for overestimated NO and HONO emissions may be a potential overestimation of active time of biocrusts by LiBry. However, the model tends to rather underestimate active time at the local scale, since it only considers the water reservoir of the photoautotrophic organisms, which is smaller than that of the whole biocrust. Hence, we think that a significant overestimation of active time by LiBry is unlikely. Finally, the relations between water saturation and NO and HONO emissions which we use in the model may not reflect the true dependence of emissions on saturation and biocrust type. The relations are obtained by averaging the saturation-emission curves of four different samples of each biocrust type, which partly show considerable variation. The sensitivity analysis demonstrates that varying the relation between water saturation and NO and HONO emissions of biocrusts by one standard deviation leads to relatively large changes in the estimated NO and HONO emissions. Also, replacing the default relation by an alternative one which is based on measurements from a different field site significantly affects simulated NO and HONO emissions by biocrusts. Further analyses of the dependence of NO and HONO emissions on the water saturation state of biocrusts are needed to determine the causes of this large variation. A potential first step in this direction would be to determine the dynamic nitrogen content of biocrusts and also of the underlying soil. Subsequently, the nitrogen content could be related to variation in the relationship between NO / HONO emissions and biocrust type. It is likely that emissions will increase with soil nitrogen content, which means that biocrusts will contribute relatively more to total emissions in areas with nitrogen-poor soils, and less on nitrogen-rich soils. Including this factor in our modeling approach would lead to a more differentiated global pattern of NO and HONO emissions. Furthermore, processes which control the non-linear dependence of NO and HONO emissions on water saturation need to be clarified. One possible explanation for the decrease of emissions at high water saturation is the limitation of the microbes which produce NO and HONO by increasingly low oxygen supply. Differences in the structure between biocrust types and, consequently, their diffusivity for oxygen, may then explain variation in the shape of the relation between water saturation and NO / HONO emissions. A fully process-based scheme of NO and HONO emissions by biocrusts may contribute to further quantifying nitrogen cycling in dryland soils. Currently, it is still difficult to close the nitrogen balance for these areas due to uncertainty regarding the amount of both nitrogen inputs via atmospheric deposition and biotic fixation and also outputs through various gaseous losses, leaching and erosion. Further analyses of the dependence of NO and HONO emissions on the water saturation state of biocrusts are needed to determine the causes of this large variation.

The sensitivity analysis shows that NO and HONO emissions seem to be inversely related to their $Q_{10}$-value. This can be explained by the fact that emissions which take place below the reference temperature of the $Q_{10}$-relationship ($25° C$) are increased if the $Q_{10}$-value decreases. Since large areas covered by biocrusts in the model are located in relatively cool regions, increased emissions there overrule the decreased emissions in warm regions at low a $Q_{10}$-value.

Due to the paucity of studies which report observation-based estimates of NO and HONO emissions by biocrusts, the validation of our model is relatively limited in this regard. To obtain annual values for NO and HONO emissions, usually

short-term measurements on wetted biocrust samples are extrapolated to the whole year based on the number of wetting events at the respective field site. Hence, for a future, more detailed validation of the LiBry model it would be useful to create data sets which include both climate data with high temporal resolution and also NO and HONO emissions from the same field site. In addition, activity of the biocrusts should be monitored, so potential mismatches between model and observations can be traced to either the simulation of the dynamic water content or the specific emissions per crust type.

Our estimates confirm the main result found by Weber et al. (2015), namely a considerable impact of biocrusts on global NO and HONO emissions. According to our simulation, biocrusts in drylands contribute around 20 % to global terrestrial NO and HONO emissions in natural ecosystems (Ciais et al., 2013). This suggests that biocrusts are important contributors to the global nitrogen cycle and have a considerable effect on chemical processes in the atmosphere. Given their biogeochemical significance, it would be interesting to assess impacts of future climate change on biocrusts and their functions. Strong decreases in NO and HONO emissions by biocrusts, which may result from reduced cover or active time, may have considerable effects on atmospheric chemistry. If climate change leads to a shift in relative cover of biocrust types towards biocrusts which release large amounts of NO and HONO, however, emissions may further increase. The LiBry model is an appropriate tool for such an assessment due to its mechanistic design, which allows direct validation of individual processes and parameters. Another interesting topic for future work related to NO and HONO emissions by biocrusts would be a more detailed analysis of the processes which cause the release of NO and HONO. This may help to explain the large variation in emissions between different samples of the same biocrust type and further constrain model-based estimates.

## 5 Conclusions

We estimated global patterns and total annual values of NO and HONO emissions by biological soil crusts (biocrusts) using the process-based model of non-vascular vegetation LiBry. We found emissions of 1.04 Tg yr$^{-1}$ NO-N and 0.69 Tg yr$^{-1}$ HONO-N by biocrusts, which corresponds to a large contribution of around 20 % of global emissions of these trace gases from natural ecosystems. This suggests a considerable impact of biocrusts on global atmospheric chemistry. Our global estimate is consistent with an earlier, empirical approach, although we estimate higher per-area emissions, which are compensated by a smaller predicted global biocrust area. Uncertainty in the empirical approach mainly resulted from assumptions concerning the frequency of metabolic activity of biocrusts in the field at the global scale. In our approach, we found a low sensitivity of NO and HONO emissions to several uncertain parameters, which are likely to affect the simulated emissions. We discuss potential approaches to improve the reliability of our estimates and we conclude that a detailed analysis of the processes which relate NO and HONO emissions to water saturation of biocrusts would be helpful in this regard.

*Code availability.* The non-vascular vegetation model LiBry used in this study is integrated in an interface for parallel computing which was developed at the Max Planck Institute for Biogeochemistry, Jena, Germany. The LiBry model excluding this interface is freely available provided that the names of the copyright holders and a disclaimer are distributed along with the code in source or binary form. The code is

available from the corresponding author upon request. Model output data which are presented as maps in this study are available as netCDF files from the authors on request.

## Appendix A: Complementary model validation figures and tables

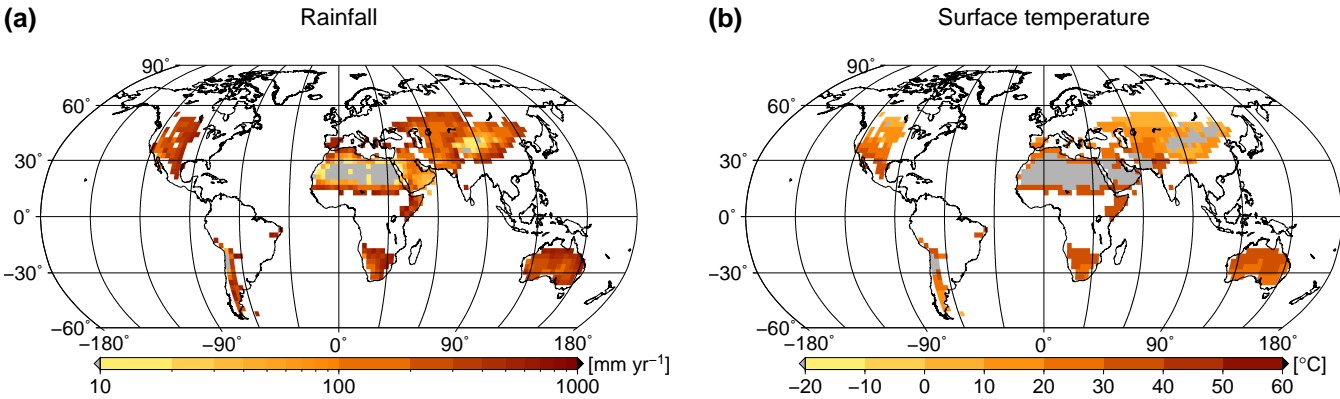

**Figure A1.** Global patterns of (a) rainfall and (b) surface temperature of biocrusts simulated by LiBry, in units of mm yr$^{-1}$ and °C, respectively. White areas denote excluded regions, while grey areas correspond to regions where no simulated strategies are able to survive.

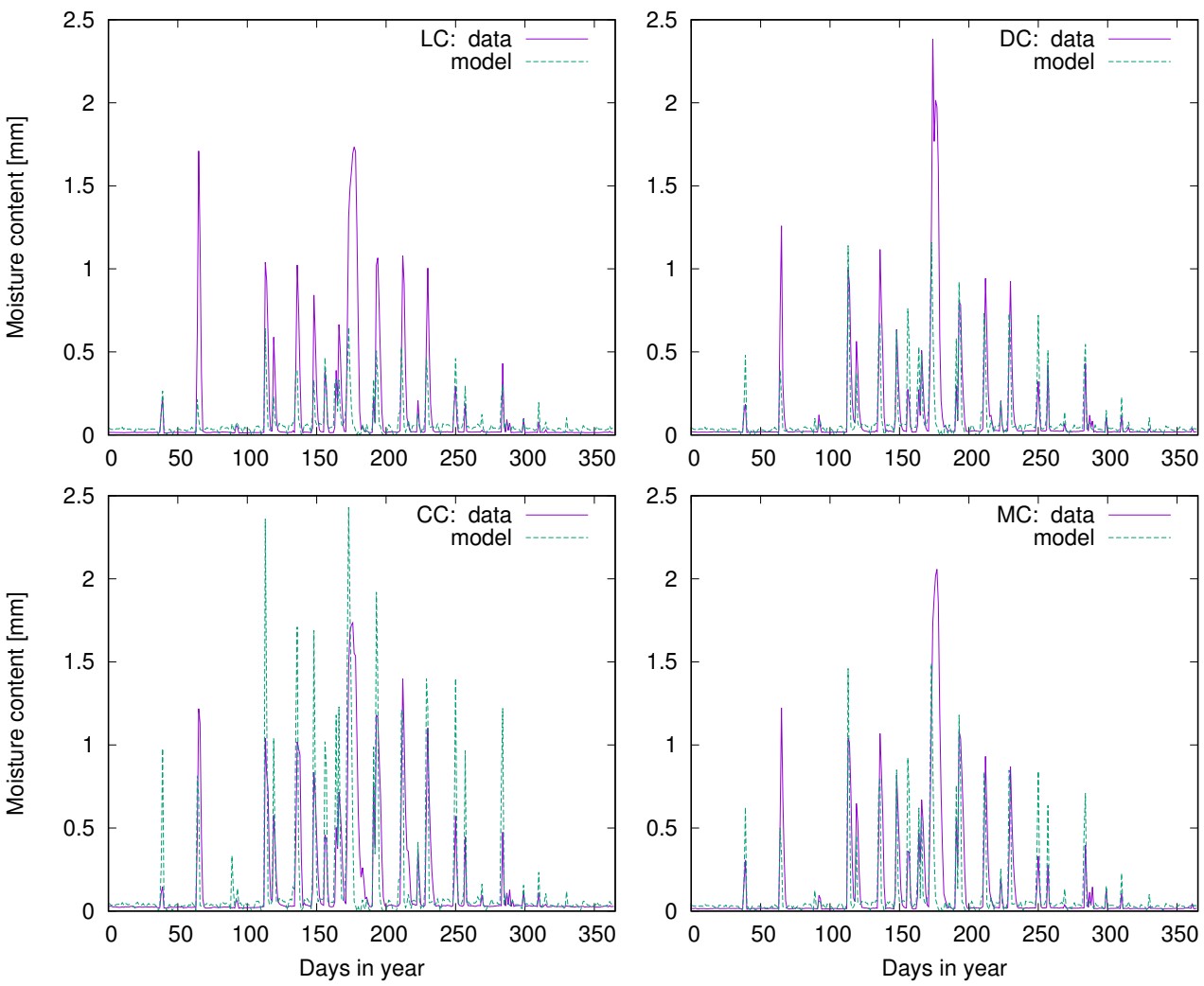

**Figure A2.** Comparison of measured and simulated moisture content for one field site and four biocrust types at Soebatsfontein. The abbreviations in the figure legend stand for (LC) light cyanobacteria-, (DC) dark cyanobacteria-, (CC) chlorolichen-, (MC) moss-dominated biocrusts.

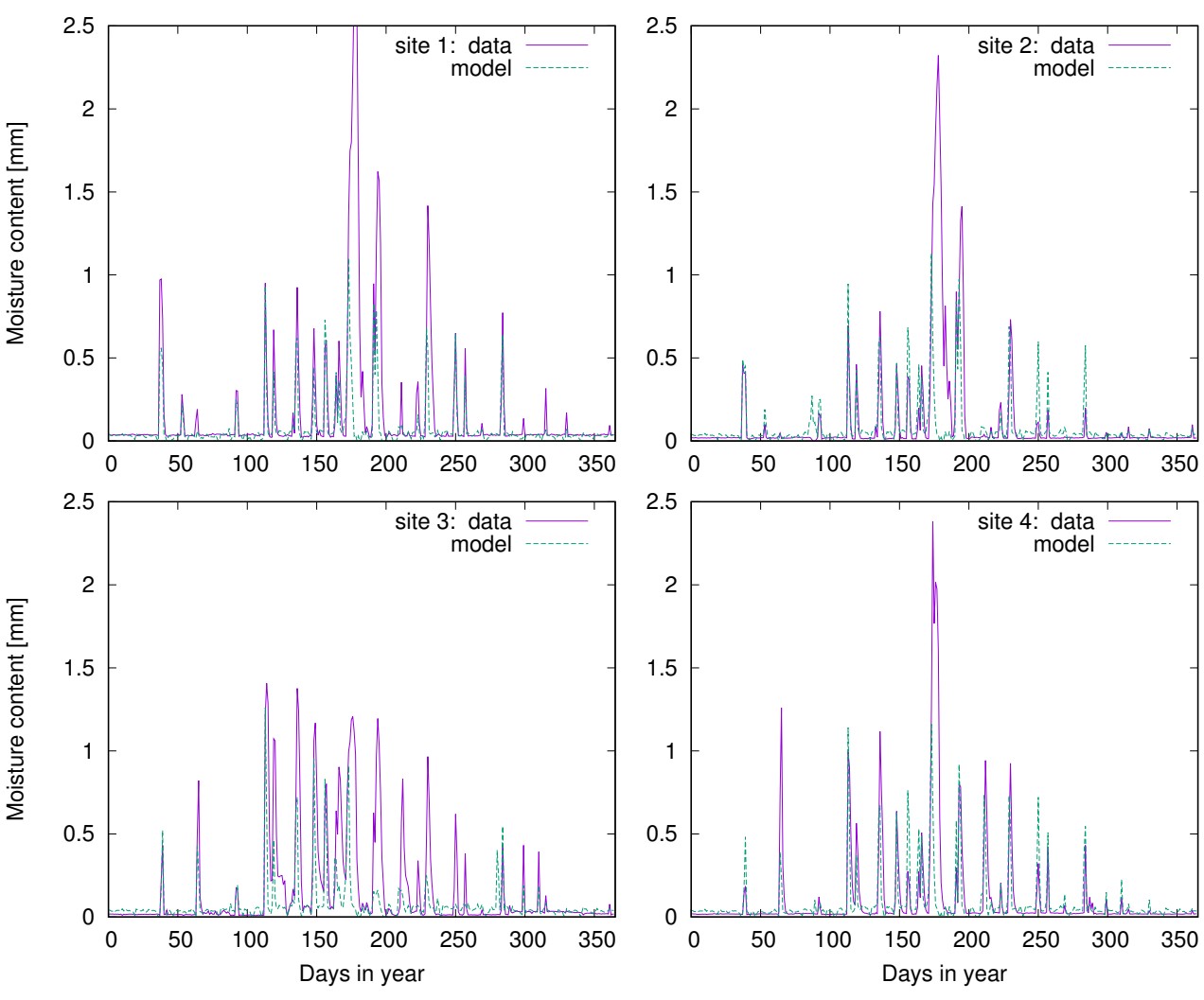

**Figure A3.** Comparison of measured and simulated moisture content for one biocrust type (dark cyanobacteria-dominated biocrust) and four field sites at Soebatsfontein.

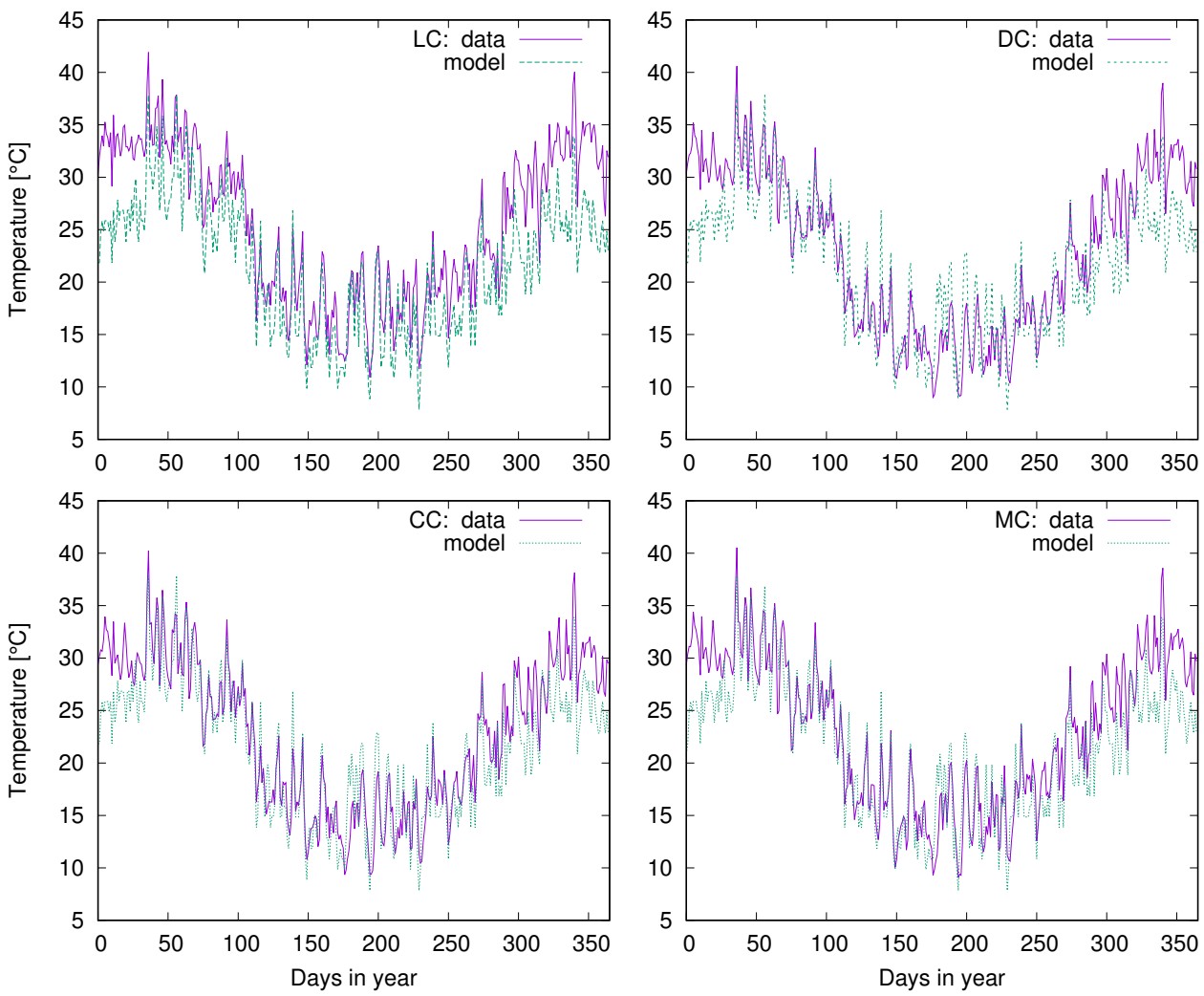

**Figure A4.** Comparison of measured and simulated surface temperature for one field site and four biocrust types at Soebatsfontein. The abbreviations in the figure legend stand for (LC) light cyanobacteria-, (DC) dark cyanobacteria-, (CC) chlorolichen-, (MC) moss-dominated biocrusts.

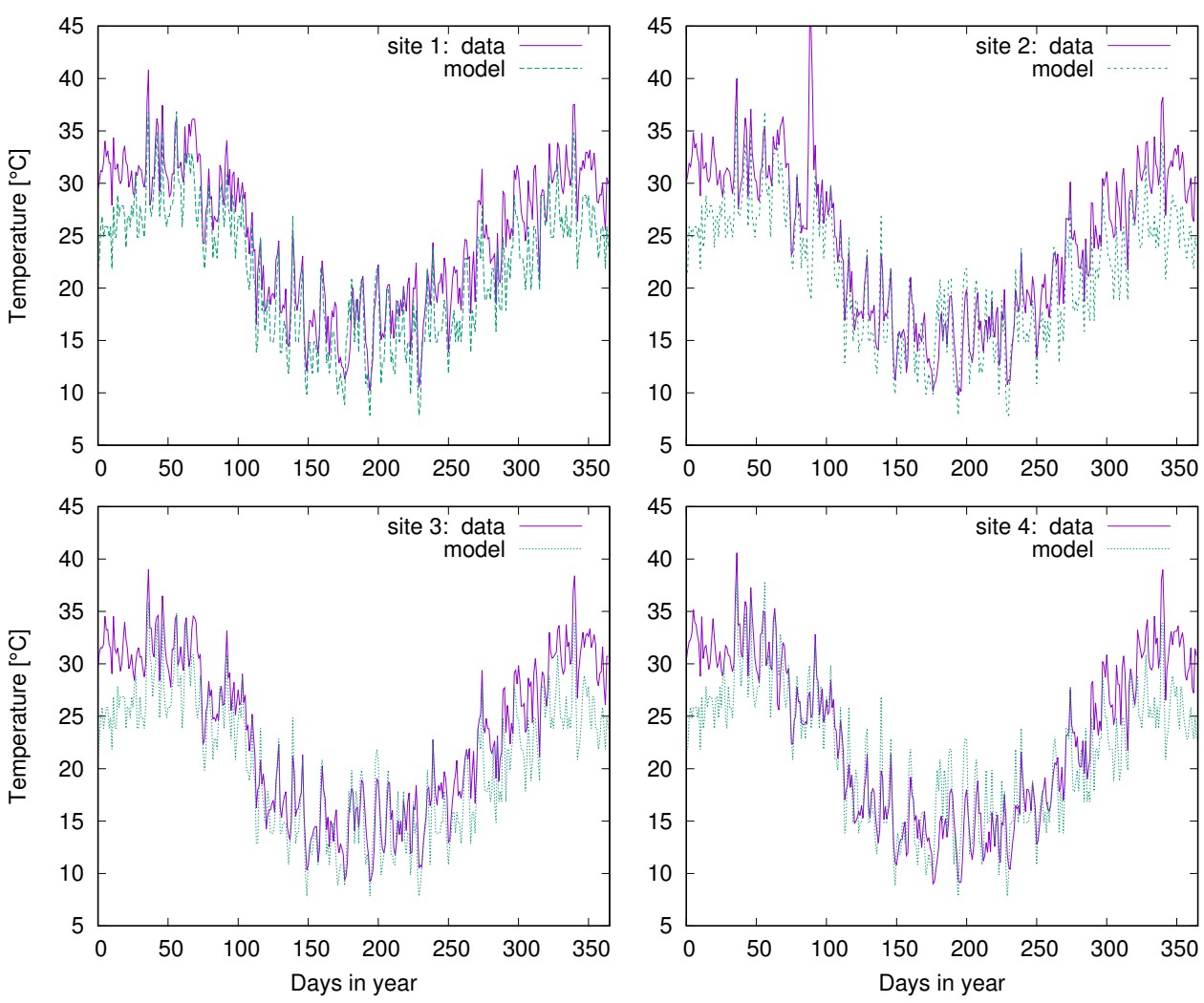

**Figure A5.** Comparison of measured and simulated surface temperature for one biocrust type (dark cyanobacteria-dominated biocrust) and four field sites at Soebatsfontein.

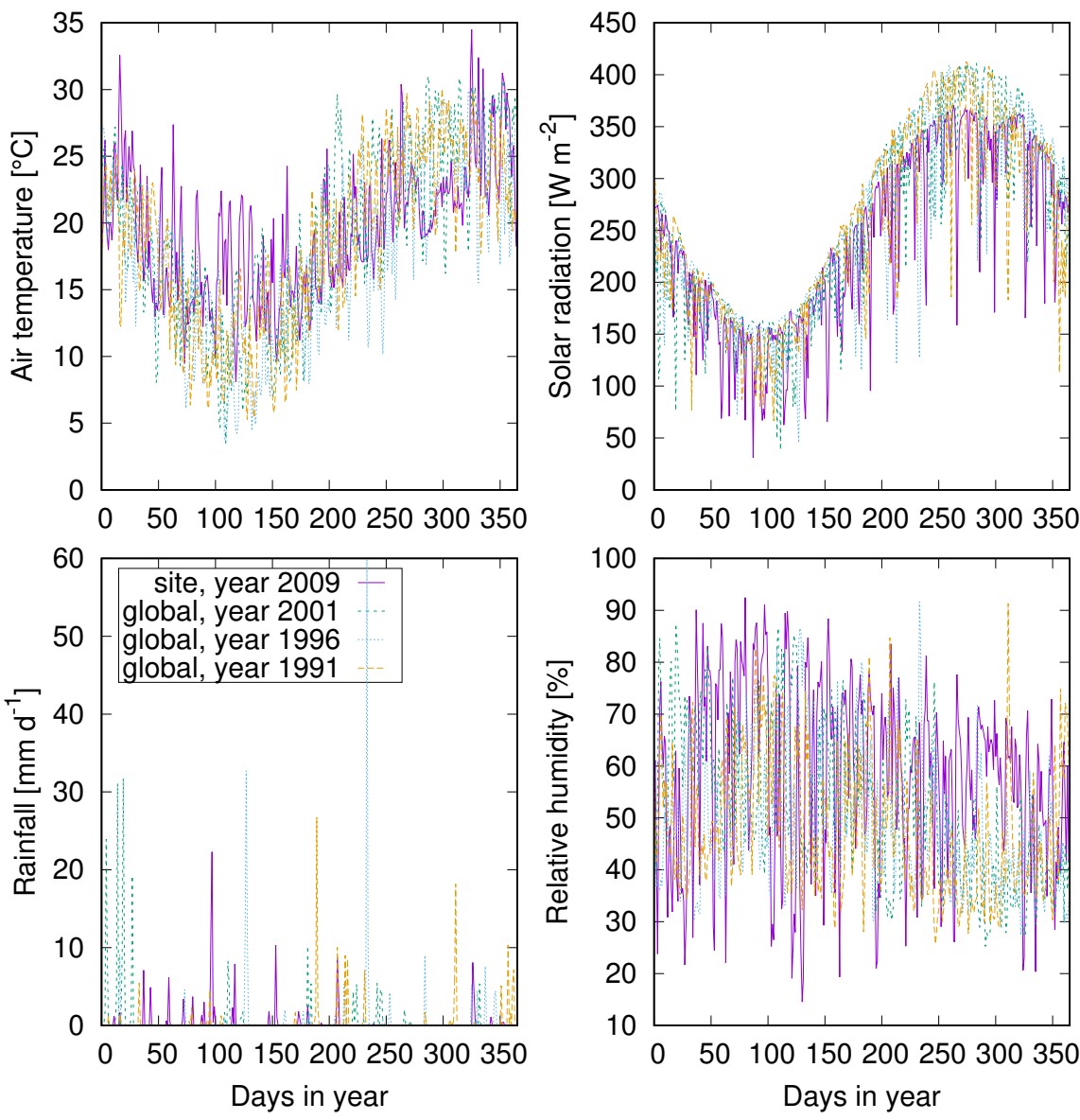

**Figure A6.** Comparison of climate data measured at one of the four Soebatsfontein field sites to global climate data for the region including Soebatsfontein. Note that global data for the year 2009 was not available from the standard data set used for LiBry, which spans the years 1958 to 2001. Climate data for the other three field sites at Soebatsfontein is similar to the site shown here.

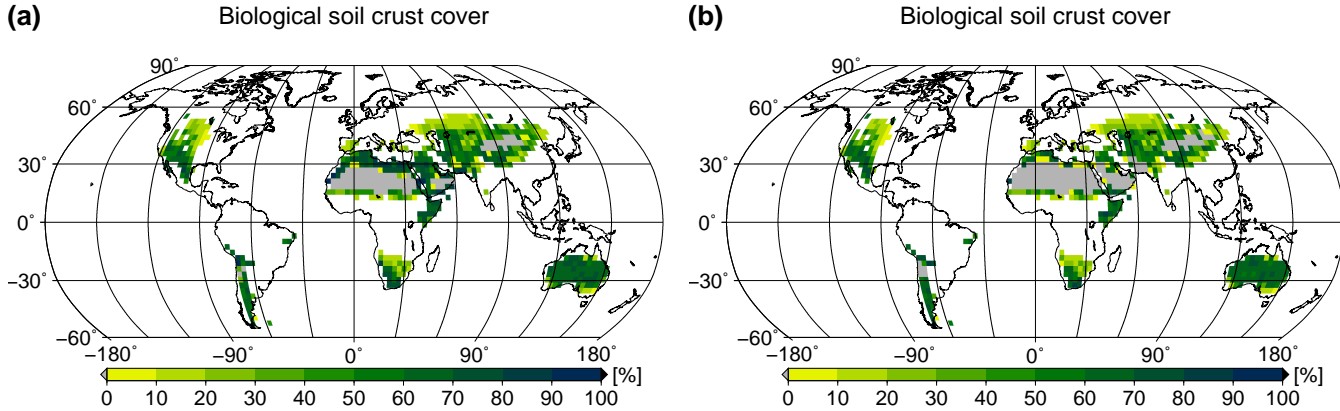

**Figure A7.** Total biocrust surface cover for (a) doubled dew availability (b) increased surface resistance to evaporation (from 0 to $100\,\mathrm{s\,m^{-1}}$).

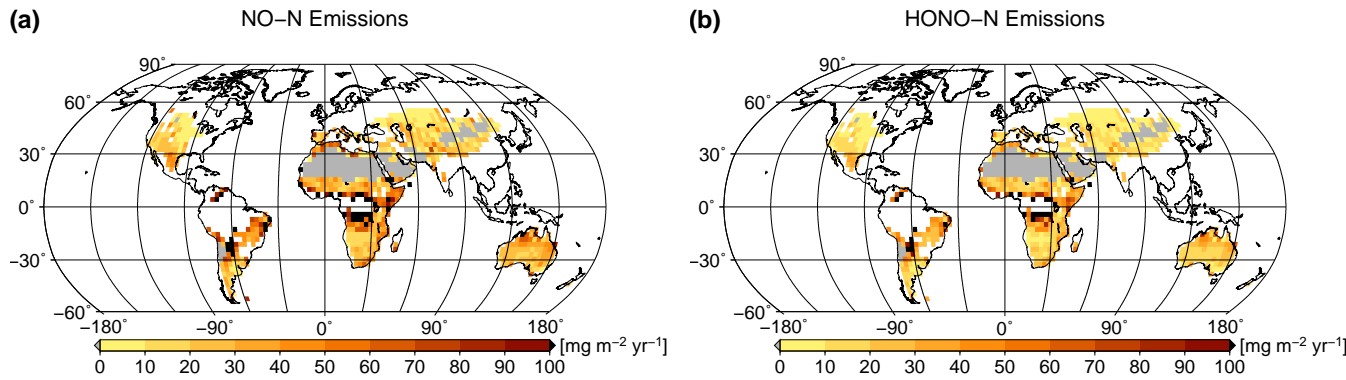

**Figure A8.** Global patterns of (a) NO and (b) HONO emissions simulated by LiBry, in units of $\mathrm{mg\,m^{-2}\,yr^{-1}}$ of nitrogen. White areas denote excluded regions, while grey areas correspond to regions where no simulated strategies are able to survive. Contrary to the standard model setup, the study region is only constrained by biome type, no maximum amount of annual rainfall is considered. Total global NO and HONO emissions by biocrusts amount to $2.48\,\mathrm{Tg\,yr^{-1}}$ NO-N and $1.69\,\mathrm{Tg\,yr^{-1}}$ HONO-N.

| Cover [%] | biome | reference |
|---|---|---|
| 100 | desert | (Karnieli and Tsoar, 1995) |
| 95 | steppe/savanna | (Xiao et al., 2010) |
| 50 | | |
| 90 | steppe/savanna | (Thomas and Dougill, 2007) |
| 85 | desert | (Lange et al., 1992) |
| 80 | desert | (Orlovsky et al., 2004) |
| 50 | desert | (Lange et al., 1990) |
| 62 | desert | (Dojani et al., 2011) |
| 60 | desert | (Lange et al., 1998) |
| 60 | desert | (Lange et al., 1997) |
| 55 | woodland | (Leys and Eldridge, 1998) |
| 37 | | |
| 52 | woodland | (Eldridge et al., 2006) |
| 43 | | |
| 35 | | |
| 50 | desert | (Brostoff, 2002) |
| 38 | steppe/savanna | (Kleiner and Harper, 1972) |
| 5 | | |
| 29 | desert | (Guo et al., 2008) |
| 53 | | |
| 29 | desert | (Zhang et al., 2007) |
| 26 | woodland | (O'Neill, 1994) |
| 21 | | |
| 25 | steppe/savanna | (Klopatek, 1992) |
| 21 | | |
| 25 | desert | (Thompson et al., 2005) |
| 21 | desert | (Jeffries and Klopatek, 1987) |
| 12 | | |
| 18 | steppe/savanna | (Graetz and Tongway, 1986) |
| 13 | desert | (Qin et al., 2006) |
| 12 | steppe/savanna | (Eldridge et al., 2000) |
| 12 | woodland | (Beymer and Klopatek, 1991) |

**Table A1.** Studies reporting values of total biocrust cover which are used in the model data comparison (Fig. 8).

| Total cover [%] | LC [%] | DC [%] | CC [%] | MC [%] | biome | reference |
|---|---|---|---|---|---|---|
| 100 | 81 | 5 | 14 | 0 | desert | (Belnap, 2002) |
| 100 | 97 | 2 | 0 | 0 | | |
| 98 | 51 | 35 | 12 | 0 | | |
| 94 | 77 | 15 | 2 | 0 | | |
| 93 | 53 | 27 | 13 | 0 | | |
| 81 | 61 | 10 | 10 | 0 | | |
| 42 | 29 | 13 | 0 | 0 | | |
| 90 | 35 | 35 | 0 | 20 | steppe/savanna | (Zhao and Xu, 2013) |
| 85 | 5 | 5 | 0 | 75 | | |
| 77 | 39 | 33 | 1 | 4 | desert | (Weber et al., 2015) |
| 77 | 26 | 45 | 3 | 3 | | |
| 72 | 40 | 31 | 1 | 0 | | |
| 64 | 34 | 29 | 0 | 1 | | |
| 76 | 62 | 0 | 4 | 10 | steppe/savanna | (Concostrina-Zubiri et al., 2014) |
| 30 | 28 | 0 | 2 | 0 | | |
| 29 | 25 | 0 | 4 | 0 | | |
| 28 | 20 | 0 | 8 | 0 | | |
| 18 | 10 | 0 | 7 | 1 | | |
| 13 | 8 | 0 | 5 | 0 | | |
| 56 | 12 | 11 | 8 | 25 | steppe/savanna | (Elseroad et al., 2010) |
| 53 | 11 | 11 | 8 | 23 | steppe/savanna | (Liu et al., 2009) |
| 46 | 13 | 9 | 4 | 20 | | |
| 42 | 8 | 9 | 7 | 18 | | |
| 36 | 8 | 7 | 13 | 8 | | |
| 24 | 1 | 7 | 14 | 2 | | |
| 20 | 0 | 6 | 13 | 1 | | |
| 31 | 14 | 13 | 4 | 0 | desert | (Belnap et al., 2014) |
| 20 | 8 | 7 | 5 | 0 | | |
| 18 | 9 | 8 | 1 | 0 | | |
| 27 | 0 | 16 | 11 | 0 | desert | (Rodríguez-Caballero et al., 2014) |
| 26 | 0 | 0 | 9 | 17 | steppe/savanna | (Dettweiler-Robinson et al., 2013) |
| 10 | 0 | 0 | 5 | 5 | | |
| 26 | 18 | 0 | 6 | 2 | steppe/savanna | (Wong et al., 2010) |
| 20 | 0 | 15 | 0 | 5 | desert | (Gómez et al., 2012) |
| 15 | 6 | 1 | 6 | 2 | desert | (Pietrasiak et al., 2014) |
| 13 | 12 | 0 | 1 | 0 | | |
| 8 | 7 | 0 | 1 | 0 | | |
| 8 | 8 | 0 | 0 | 0 | | |
| 7 | 5 | 0 | 2 | 0 | | |
| 3 | 2 | 0 | 1 | 0 | | |

**Table A2.** Studies reporting values of total biocrust cover and relative cover of four biocrust types which are used in the model data comparison (Fig. 8). The abbreviations stand for (LC) light cyanobacteria-, (DC) dark cyanobacteria-, (CC) chlorolichen-, (MC) moss-dominated biocrusts. All cover values are in percent.

*Competing interests.* The authors declare that they have no competing interests.

*Acknowledgements.* We would like to thank all the authors of the initial study on NO- and HONO-emissions of biological soil crusts(Weber et al., 2015), which provided input data for our current investigation. BW would like to thank Paul Crutzen for awarding her the Nobel Laureate Fellowship and PP gladly appreciates funding by the Deutsche Forschungsgemeinschaft (DFG, German Research Foundation) -
5   408092731. The Max Planck Institute for Biogeochemistry provided computational resources.

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
