# Peer review of "Global NO and HONO emissions of biological soil crusts estimated by a process-based non-vascular vegetation model"

_Biogeosciences, 2018_

## Referee Comment (RC1) · Anonymous Referee #1 · 27 Oct 2018

General comments: This study provided an estimate of global NO and HONO emissions from biocrusts by improving a non-vascular vegetation model (LiBry). Specifically, the authors improved the model to predict relative cover of different biocrust types and then estimated NO and HONO emissions through extrapolating the relationship between NO and HONO emissions and water content (i.e. Figure 1). While the authors did some interesting works, I think there have a couple of major deficiencies in this study. First, the authors did not justify the worldwide applicability of the relationship between NO and HONO emissions and water content while this relationship is critical for the products (global NO and HONO emissions) presented in this manuscript. Second, the authors did very limited model validation for the NO and HONO emissions. These

deficiencies make me feel the major products, global NO and HONO emissions, are not very solid in the current version of the manuscript. Therefore, I think substantial works/improvements are necessary for addressing the deficiencies I mentioned.

Other specific points: Page 2 Line 19: Grammar error.

Page 2 Lines 33 to 34: Grammar error in this sentence.

Page 4 Line 29: Add a full stop.

Page 5 Line 15: How reliable of this relationship/method? How about applicability and uncertainties of this relationship/method?

Page 9 Line 26: Past tense should be used for what you did. Please make necessary changes throughout the MS.

Page 11 Lines 26 to 27: This sentence is not clear.

Page 15 Line 1: The sub-titles of the result section are confusing. You described a lot of results before '3.1 Global validation'; so what is the sub-title for the results described before 3.1?

Page 15 Line 15: 'report' should be 'reported'. This issue has been found throughout the manuscript.

Page 18 Line 14: Is the low sensitivity due to the simple model for NO and HONO emissions?

Page 20 Line 30 to 33: It might be worthy to provide some explanations for this statement. Why this method is more appropriate than calibration?

Page 21 Line 19: 'that' should be 'than'.

Page 22 Line 30 to 33: Does NO and HONO emissions affected by other factors in bicrust systems, such as N content etc?

---

## Referee Comment (RC2) · Anonymous Referee #2 · 21 Dec 2018

The Porada et al. article uses a process-based modeling approach with the LiBry model to determine the global areal coverage of biocrust types responsible for emitting NO and HONO, which are important atmospheric chemical constituents for OH. They then relate biocrust type to water saturation to determine emissions of these two constituents. The paper is well written, has sufficient validation and sensitivity analysis, and concludes with values similar to an empirical upscaling study, but for a more restrained area, so that, in fact, the emissions are actually even larger. The wetting event validation comes from four sites in a single South African location, so would be good if there was an attempt for more global validation of this variable, as it is key to how emission are modeled. There are quite a few things that can be revised to help clarify

the paper, as suggested below.

1. The four biocrust types are light and dark cyanobacteria, chlorolichen, and mosses, yet the LiBry model is Lichen and Bryophyte. At times bryophyte is used instead of mosses (p. 20 line 8), so should just specify that up front and then use "mosses" consistently throughout. 2. P. 2, line 19 – 100% of N2O emissions in dryland regions is what percent of total N2O emissions? 3. More explanation of physiological strategies in LiBry. A few of them are used to partition between the four types as in Figure 2. But in section 2.4 (first sentence) it is mentioned that there 3000 physiological strategies, so it is not clear what all these strategies are referring to. Also, some further explanation of how photosynthetic capacity, height, and CO2 diffusivity are used in the model would be helpful. 4. At end of Introduction there are two main extensions of LiBry mentioned – assigning physiological strategies to the four types (so, is this really just defining new types by physiological strategies – would make it clearer to describe it in this way); the dynamic surface cover model described in Figure 3; and how about adding determining emissions of NO and HONO to the model? 5. The fundamental assumptions from Figure 1 that underlie the emissions model are pretty simplistic. So, how about further discussion of what controls water saturation – not just in terms of water balance, but how important is the uptake of water by the photosynthetic biocrusts? 6. NO and HONO emission are based on water saturation and Q10 – is it simply these two terms multiplied? Include an equation that explicitly states how it is calculated. 7. Since the effect of nitrogen cycling is not included (p. 5, lines 16-17), there should be some discussion of that in the Discussion as to how that may change things. 8. Figure 3: I suggest changing the green colors so they grade from darker to lighter with height, rather than the lightest one in the middle. 9. Need reference for the WATCH data (p. 8, line 13) 10. I would like to see more explanation of how disturbance is applied (p. 9, line 6) – although the Porada reference is given, one or two sentences here would be helpful. 11. Table A2: list units as %. 12. Sensitivity to model parameters (end of Methods) is based on those that affect total biocrust cover, relative cover, temperature-dependence of NO and

HONO emissions, but what about those that affect water saturation? There ought to be some test of a key parameter here, as that affects NO and HONO emissions. 13. P. 11, line 2 – why are chlorolichen-dominated crusts larger fractions of the Sahara and Arabian deserts – does not look that way from the figure? 14. From Figure 4e, I would conclude the light cyanobacteria are the outlier under low precipitation, rather than lumping the dark cyanobacteria together with them, which is done throughout the text. 15. Appendix figures are out of order – A6 is discussed first, then A2-A5, then A1, and finally A7. Please put these in order. Also, from Figure 9 and Figure A2-A5 I am unclear which site (1 - 4) is being referred to, so there should be some way of distinguishing between the four sites. Furthermore, why are the results from only one of these sites shown in the main text? 16. In section 4.1, there should be better attribution of these explanations as to what actually occurs in the model, to distinguish from general arguments. For example, why is there a competitive advantage due to height given that shading is not an issue? How does the discussion of moss vs lichen in dry/wet conditions pertain to the model design? Also, p. 20 line 4 – not sure how this follows previous sentence. 17. P. 20, line 34 – it is ok for process-based models to determine parameters based on optimization from field results – do not need to just use values based on the literature. 18. P. 21, line 5: Are the results not sensitive to climate or is it just that at this one site the match is good? 19. P. 21, line 23: Would changing the albedo throw off the other seasons, or is it a change that only occurs in the warm season? 20. P. 22, line 23: The conclusion of correct wetting events is from only the one location in South Africa, at least as presented in this paper. There really needs to be a more broad-scale analysis globally of this in order to make this conclusion. Is that possible to do – as that is really key to getting the emissions correct? 21. I would consider putting the 1.1 (which is listed as 1.04 in the abstract?) and 0.6 Tg/yr into the Conclusions and adding the 20% in the Abstract.

Please also note the supplement to this comment:
https://www.biogeosciences-discuss.net/bg-2018-324/bg-2018-324-RC2-

supplement.pdf

---

## Author Comment (AC1) · 9 Feb 2019

**Global NO and HONO emissions of biological soil crusts estimated by a process-based non-vascular vegetation model**

Philipp Porada1,2, Alexandra Tamm2, Jose Raggio3, Axel Kleidon4, Ulrich Pöschl2, and Bettina Weber2

1University of Potsdam, Vegetation Ecology and Nature Conservation, Am Mühlenberg 3, 14476 Potsdam, Germany

2Max Planck Institute for Chemistry, Multiphase Chemistry Department, P.O. Box 3060, 55020 Mainz, Germany

3Departamento de Biología Vegetal II, Facultad de Farmacia, Universidad Complutense de Madrid, Madrid, Spain

4Max Planck Institute for Biogeochemistry, P.O. Box 10 01 64, 07701 Jena, Germany

We thank the referees for their useful and thorough comments which will be helpful to improve our manuscript. We have prepared a detailed response to the points raised by the referees. We show the referees' comments in italic text, while our responses are formatted as standard text.

**Response to the comments of referee #1**

General comments: This study provided an estimate of global NO and HONO emissions from biocrusts by improving a non-vascular vegetation model (LiBry). Specifically, the authors improved the model to predict relative cover of different biocrust types and then estimated NO and HONO emissions through extrapolating the relationship between NO and HONO emissions and water content (i.e. Figure 1). While the authors did some interesting works, I think there have a couple of major deficiencies in this study. First, the authors did not justify the worldwide applicability of the relationship between NO and HONO emissions and water content while this relationship is critical for the products (global NO and HONO emissions) presented in this manuscript. Second, the authors did very limited model validation for the NO and HONO emissions. These deficiencies make me feel the major products, global NO and HONO emissions, are not very solid in the current version of the manuscript. Therefore, I think substantial works/improvements are necessary for addressing the deficiencies I mentioned.

We are glad that the referee thinks that our study is interesting. We agree that the uncertainty associated with our estimate on NO and HONO emissions could be better analysed and constrained. In the revised manuscript, we will point out more clearly the uncertainty of our estimate. We will extend the abstract (before the last sentence):

"Based on this, we estimate global total values of  $1.04 \text{ Tg yr}^{-1}$  NO-N and  $0.69 \text{ Tg yr}^{-1}$  HONO-N released by biological soil crusts. Due to the low number of observations on NO and HONO emissions suitable to validate the model, our estimates are still relatively uncertain. However, they are consistent with the amount estimated by the empirical approach, which confirms that biological soil crusts are likely to have a strong impact on global atmospheric chemistry via emissions of NO and HONO."

Moreover, we found one additional study which reports NO emissions by biocrusts, which we will add to the revised manuscript (P 15 L 23):

"Barger et al. [2005] estimate annual NO emissions of 2 to 16 mg m-2 yr-1 by biological soil crusts at two fields sites in the Canyonlands National Park, Utah, USA. This compares well to the large-scale estimate of 10 to 20 mg m-2 yr-1 of NO-N simulated by LiBry for this region."

In the revised discussion, we will add (P 23 L 2):

"Due to the paucity of studies which report observation-based estimates of NO and HONO emissions by biological soil crusts, the validation of our model is relatively limited in this regard. To obtain annual values for NO and HONO emissions, usually short-term measurements on wetted biological soil crust samples are extrapolated to the whole year based on the number of wetting events at the respective field site. Hence, for a future, more detailed validation of the LiBry model it would be useful to create data sets which include both climate data with high temporal resolution and also NO and HONO emissions from the same field site. In addition, activity of the biocrusts should be monitored, so potential mismatches between model and observations can be traced to either the simulation of the dynamic water content or the specific emissions per crust type."

To better quantify the uncertainty of our estimate, we carried out additional sensitivity analyses on the relationship between water saturation of biological soil crusts and their NO and HONO emissions. These will be described as follows in the Methods section of the revised manuscript (P 10 L 20):

"Moreover, we account for uncertainty resulting from variation in the measured relations between water saturation and NO and HONO emissions of different biocrust types.

(f) We calculated the standard deviation of the measurements made by [Weber et al., 2015], and subtracted it from the average curves shown in Fig. 1 to create a lower bound of NO and HONO emissions as a function of biocrust water content. To create a corresponding upper bound, we added one standard deviation to the curves.

(g) We replaced our default relationship between water content and NO and HONO emissions of different biocrust types by an alternative one etablished by Meusel et al. [2018]. They used a similar approach, but for a different location, a field site in Cyprus."

In the Results section, we will add the following (P 18 L 14, see also extended Tab. 2 below at point 12 of referee #2):

"Simulated NO and HONO emissions are also sensitive to variation in the relationship between water content and emissions. Decreasing the specific emissions at a given water content by one standard deviation reduces simulated total global NO emissions by 25% and HONO emissions by 50% compared to the originally estimated emissions (see Tab. 2). Increasing specific emissions by one standard deviation raises simulated NO emissions by 85% and HONO emissions by 107%. Moreover, replacing our default relation between water content and NO and HONO emissions, which is based on Weber et al. [2015], by an alternative relation derived from Meusel et al. [2018], significantly affects our estimates. While simulated NO emissions decrease by 66%, HONO emissions remain almost unchanged and are reduced by only 3%."

We will also update the Discussion with the following text (P 22 L 31):

"The sensitivity analysis demonstrates that varying the relation between water saturation and NO and HONO emissions of biocrusts by one standard deviation leads to relatively large changes in the estimated NO and HONO emissions. Also, replacing the default relation by an alternative one which is based on measurements from a different field site significantly affects simulated NO and HONO emissions by biocrusts. Further analyses of the dependence of NO and HONO emissions on the water saturation state of biocrusts are needed to determine the causes of this large variation. A potential first step in this direction would be to determine the dynamic nitrogen content of biocrusts and also of the underlying soil. Subsequently, the nitrogen content could be related to variation in the relationship between NO/HONO emissions and biocrust type. It is likely that emissions will increase with soil nitrogen content, which means that biocrusts will contribute relatively more to total emissions in areas with nitrogen-poor soils, and less on nitrogen-rich soils. Including this factor in our modeling approach would lead to a more differentiated global pattern of NO and HONO emissions. Furthermore, processes which control the non-linear dependence of NO and HONO emissions on water saturation need to be clarified. One possible explanation for the decrease of emissions at high water saturation is the limitation of the microbes which produce NO and HONO by increasingly low oxygen supply. Differences in the structure between biocrust types and, consequently, their diffusivity for oxygen, may then explain variation in the shape of the relation between water saturation and NO / HONO emissions. A fully process-based scheme of NO and HONO emissions by biocrusts may contribute to further quantifying nitrogen cycling in dryland soils. Currently, it is still difficult to close the nitrogen balance for these areas due to uncertainty regarding the amount of both nitrogen inputs via atmospheric deposition and biotic fixation and also outputs through various gaseous losses, leaching and erosion."

Other specific points: Page 2 Line 19: Grammar error.

In the revised version, we will change this sentence (see point 2 by referee #2).

Page 2 Lines 33 to 34: Grammar error in this sentence.

We are not able to find a grammar error in this sentence, but this issue may be resolved during the copy-editing. Page 4 Line 29: Add a full stop.

We will add a full stop in the revised manuscript.

Page 5 Line 15: How reliable of this relationship/method? How about applicability and uncertainties of this relationship/method?

In the revised manuscript, we will discuss in more detail the potential reasons for variation in the relationship between water content and NO and HONO emissions of biocrusts. Moreover, we will quantify the sensitivity of the model estimates to variation in this relationship (see our first point above).

Page 9 Line 26: Past tense should be used for what you did. Please make necessary changes throughout the MS.

In the revised version of the manuscript, we will use past tense for the description of our work.

Page 11 Lines 26 to 27: This sentence is not clear.

We agree with the referee that this sentence can be improved and we will change it in the revised manuscript as follows:

"The large-scale spatial pattern of active time shows highest values in more poleward regions, medium values in large parts of (sub)tropical regions and lowest values in desert regions. This may be explained by a combination of the patterns of rainfall and surface temperature (see Fig. A6): In poleward regions, rainfall is relatively high and evaporation is moderate, due to lower surface temperatures, resulting in relatively large water supply. In many tropical regions, rainfall is also high but evaporation is markedly increased compared to more poleward regions, as indicated by the higher surface temperatures. This results in less available water and thus in less active time."

Page 15 Line 1: The sub-titles of the result section are confusing. You described a lot of results before 3.1 Global validation; so what is the sub-title for the results described before 3.1?

We will add another subtitle called "Global patterns of biocrust cover and NO and HONO emissions" for the beginning of the Results section.

Page 15 Line 15: report should be reported. This issue has been found throughout the manuscript.

We will change the revised manuscript in this regard.

Page 18 Line 14: Is the low sensitivity due to the simple model for NO and HONO emissions?

The main reason is the low sensitivity of active time to the parameters tested in the original manuscript (see Tab. 2). In the revised manuscript, we will include an extended sensitivity analysis, which also focuses on the water balance of the organisms. Here, we partly find a higher sensitivity of NO and HONO emissions, as discussed in the response to the comments of referee #2 (point 12).

Page 20 Line 30 to 33: It might be worthy to provide some explanations for this statement. Why this method is more appropriate than calibration?

We agree that this should be better explained. In the revised manuscript, we will provide a more detailed reasoning, where we argue that the physiological properties which are used to distinguish between the different biocrust types are not the only uncertain parameters in our approach. Hence, by calibrating the model with regard to these physiological properties, we would implicitly assume that all other uncertain parameters have the correct values, which is not necessarily true (see also point 17 by referee #2).

**Page 21 Line 19: that should be than.**

We will correct this in the revised manuscript.

Page 22 Line 30 to 33: Does NO and HONO emissions affected by other factors in bicrust systems, such as N content etc?

In reality, the N contents of the biocrust and also the underlying soil most likely influence the shape and magnitude of the relationship between water saturation and NO / HONO emissions. In the revised manuscript, we will extend the discussion in this regard (see our first point above):

**Response to the comments of referee #2**

The Porada et al. article uses a process-based modeling approach with the LiBry model to determine the global areal coverage of biocrust types responsible for emitting NO and HONO, which are important atmospheric chemical constituents for OH. They then relate biocrust type to water saturation to determine emissions of these two constituents. The paper is well written, has sufficient validation and sensitivity analysis, and concludes with values similar to an empirical upscaling study, but for a more restrained area, so that, in fact, the emissions are actually even larger. The wetting event validation comes from four sites in a single South African location, so would be good if there was an attempt for more global validation of this variable, as it is key to how emission are modeled. There are quite a few things that can be revised to help clarify the paper, as suggested below. We are glad that the referee makes a positive assessment on our paper, in particular regarding validation and sensitivity analysis. We agree with the referee that the manuscript would benefit from a more extended validation of the simulated dynamic water saturation. Therefore, in the revised version of our manuscript, we compare the model to observational data from another site with a different climate, which is located in South Spain, and we also carry out an extended sensitivity analysis with respect to the dynamic water content (see responses to comments 12 and 20 below for details).

1. The four biocrust types are light and dark cyanobacteria, chlorolichen, and mosses, yet the LiBry model is Lichen and Bryophyte. At times bryophyte is used instead of mosses (p. 20 line 8), so should just specify that up front and then use mosses consistently throughout.

We agree that this can be made clearer, and we will explain in the revised manuscript at the beginning of Sect. 2 that we use "mosses" here instead of "bryophytes".

2. P. 2, line 19 100% of N2O emissions in dryland regions is what percent of total N2O emissions?

We will change this sentence in the revised manuscript to the following:

"The authors found that these organisms are responsible for 4-9% of global emissions of N2O from natural terrestrial sources. In drylands, where they represent main components of biocrusts, they may even contribute up to 100% to N2O emissions."

3. More explanation of physiological strategies in LiBry. A few of them are used to partition between the four types as in Figure 2. But in section 2.4 (first sentence) it is mentioned that there 3000 physiological strategies, so it is not clear what all these strategies are referring to. Also, some further explanation of how photosynthetic capacity, height, and CO 2 diffusivity are used in the model would be helpful.

We agree with the referee that the description of the physiological strategies and their relation to different biocrust types could be more detailed. In fact, not only a few of the strategies are used to partition between the biocrust types, but all of them, which means that each of the 3000 strategies is assigned to one of the four crust types at the start of the simulation. In the revised manuscript, we will therefore extend the corresponding paragraph (P 6 L 3-10) by the following text:

"To assign strategies to biocrust types, we first determine to which group of photoautotrophs (lichens, mosses or cyanobacteria) each simulated physiological strategy belongs. This is necessary since the LiBry model does not categorize the simulated strategies by default. Instead, an individual strategy is defined only through its unique combination of values of several physiological parameters, as described above. We use these physiological parameters to distinguish the strategies into lichens, mosses and cyanobacteria. For this purpose, the following parameters are taken into account: Height, CO2- diffusivity in the wet state, and photosynthetic capacity. The growth height of a strategy has several effects in the model: For the same amount of cover expansion, the higher a strategy is, the more biomass is needed, which is a competitve disadvantage. However, taller strategies have more potential to store water per given area, and they may also outcompete smaller strategies with regard to light availability. The  $CO_2$ -diffusivity at high water saturation is an important physiological constraint, since organisms with higher diffusivity are able to grow more than those with low diffusivity in the model. This advantage is, however, associated with increased loss of water through evaporation for given climatic conditions, due to the more open structure of the biomass. Photosynthetic capacity controls the ability of a photoautotroph to use high light intensities and to capture  $CO_2$  from the atmosphere. Strategies with a high photosynthetic capacity are able to grow more than those with low capacity under certain climatic conditions, but this advantage comes at the cost of increased maintenance respiration and turnover. We want to mention that the categorization of strategies into lichens, mosses and cyanobacteria has no impact on the dynamics of the vegetation in the model, it only affects the simulated NO and HONO emissions."

4. At end of Introduction there are two main extensions of LiBry mentioned - assigning physiological strategies to the four types (so, is this really just defining new types by physiological strategies - would make it clearer to describe it in this way); the dynamic surface cover model described in Figure 3; and how about adding determining emissions of NO and HONO to the model?

We agree that these points should be clarified and we will change the end of the Introduction (P 3 L 24) accordingly:

"... To this end, we extend the LiBry model in three central aspects: First, we introduce a scheme which categorizes the large number of physiological strategies simulated by LiBry for drylands into lichens, mosses and cyanobacteria. We then define the different biocrust types considered in the study by Weber et al. [2015] according to these vegetation groups. This enables us to take into account the strong differences in NO and HONO emissions between biocrust types. Secondly, we alter the scheme for dynamic surface cover of the physiological strategies in LiBry, which enables us to predict the relative cover of each biocrust type. Thirdly, we extend LiBry by an empirical scheme which calculates NO and HONO emissions of different biocrust types based on their water saturation. Thereby, saturation of the biocrusts is based on the dynamic water content of the individual physiological strategies simulated by LiBry. We evaluate our estimates of biocrust surface cover both at the local and the global scale by comparison to observations and we compare simulated NO and HONO emissions to the available estimates from the literature."

5. The fundamental assumptions from Figure 1 that underlie the emissions model are pretty simplistic. So, how about further discussion of what controls water saturation - not just in terms of water balance, but how important is the uptake of water by the photosynthetic biocrusts? We agree with the referee that it is useful to describe water relations in more detail and we will therefore extend the model description by the following text (P 4 L 16):

"Several physiological properties regulate the dynamic water saturation of non-vascular vegetation in the model: First, the uptake of water is limited not only by rainfall or snow melt, but also by the water storage capacity of the organisms, which depends on it height and the porosity of the biomass. At full saturation, additional water input infiltrates into the soil. The extent to which dew can be used as a water source in the model depends mostly on climatic conditions, and to a limited extent on properties of the organisms which influence the surface temperature, e.g. via evaporation. Secondly, also water loss is regulated by properties of the organisms. These are the same as for water uptake, namely the specific water storage capacity, capillary structure of the biomass and albedo. Note that non-vascular vegetation does not possess stomata, so an active reduction of evaporation is not possible."

We will also add more explanations on how water saturation is related to NO and HONO emissions (see below).

6. NO and HONO emission are based on water saturation and Q10 - is it simply these two terms multiplied? Include an equation that explicitly states how it is calculated.

To derive NO and HONO emissions from water saturation, we created a look-up table based on a discetization of the measured curves shown in Fig.1. The values of NO and HONO emissions for a given value of water saturation were subsequently scaled as a function of temperature based on a Q10-relationship. Therefore, we cannot provide an equation for the complete calculation of the emissions. Instead, we will explain this approach in more detail and include the following text in the revised manuscript (P 5 L 15):

"To implement these relations into the LiBry model, we discretize the curves shown in Fig. 1 and create a look-up table, which assigns values of NO and HONO emissions for each value of water saturation. Subsequently, the emissions are scaled according to surface temperature:

$$E_{\rm NO,ONO} = E_{\rm NO,HONO}(\Theta) Q^{\frac{I_{\rm S} - I_{\rm REF}}{10.0}}$$
(1)

where  $E_{\text{NO,HONO}}$  are the emissions of NO and HONO, respectively,  $E_{\text{NO,HONO}}(\Theta)$  are the emissions at a given water saturation  $\Theta$  based on the look-up table, Q is the  $Q_{10}$ value, and  $T_{\text{S}} - T_{\text{REF}}$  is the difference between the surface temperature of the simulated organisms and the reference temperature. In this way, NO and HONO emissions are calculated from the simulated water saturation at each time step of the model run."

7. Since the effect of nitrogen cycling is not included (p. 5, lines 16-17), there should be some discussion of that in the Discussion as to how that may change things.

We agree that the Discussion should be extended in this regard and we will add a corresponding text to the revised manuscript (see first point of the response to referee #1).

8. Figure 3: I suggest changing the green colors so they grade from darker to lighter with height, rather than the lightest one in the middle.

We will change this accordingly in the revised manuscript.

9. Need reference for the WATCH data (p. 8, line 13)

We cite Weedon et al. [2011] in the manuscript as this is the recommended reference. To make this more clear, we will extend the reference in the following way:

"Climate forcing data are based on the WATCH data set (see Weedon et al. [2011] and also http://www.eu-watch.org/data\_availability), which span the years 1958 to 2001."

10. I would like to see more explanation of how disturbance is applied (p. 9, line 6) - although the Porada reference is given, one or two sentences here would be helpful.

We will add in the revised manuscript the following, to explain this in more detail (P 9 L 7):

"... This means, that for each of the 16 biomes which are considered in LiBry at the global scale, a characteristic interval between disturbance events is determined from the literature. Thereby, we account for processes such as fire, windbreak of trees, which destroys the habitat of epiphytic non-vascular vegetation, and also trampling by animals. The disturbance interval is then converted into the fraction of the biocrust surface cover which is destroyed once per month in the simulation, e.g an interval of 8 years would result in roughly 1% of the surface cover being disturbed each month."

11. Table A2: list units as %.

We will add the unit % to the table header.

12. Sensitivity to model parameters (end of Methods) is based on those that affect total biocrust cover, relative cover, temperature-dependence of NO and HONO emissions, but what about those that affect water saturation? There ought to be some test of a key parameter here, as that affects NO and HONO emissions.

We agree with the referee that it would be useful to extend the sensitivity analysis in this regard. Therefore, we have carried out additional tests, which we will describe in the revised manuscript in the following way (end of Methods):

[revised manuscript text omitted]

Finally, we will discuss the outcomes of the sensitivity analysis in the Discussion (P 21 L 26):

"The relatively large sensitivity of our estimated NO and HONO emissions to parameters which influence the dynamic water content is plausible. If maximum available dew is doubled, or a resistance to evaporation is introduced, the potential area where biocrusts may occur in the model is significantly larger and total biocrust cover and active time significantly increase (Tab. 2). The strong increase in NO and HONO emissions can further be explained by the large increase in biocrust cover in warm regions, where high temperatures further enhance NO and HONO release (Eq. 1). However, the simulated global patterns of biocrust coverage under doubled dew or increased surface resistance seem to be inconsistent with observations. In extremely dry regions of North Africa, Arabia and Australia, high cover values of more than 70% are simulated, which are substantially higher than large-scale average values which are commonly assumed for these regions (see Fig. A8 NEW). Moreover, although dew of 80 mm yr-1 may occur locally under certain conditions in drylands, this value is most likely too high for a large-scale estimate (see also Vuollekoski et al. [2015] for typical values),"

13. P. 11, line 2 - why are chlorolichen-dominated crusts larger fractions of the Sahara and Arabian deserts - does not look that way from the figure?

We actually wanted to say that chlorolichen-dominated crusts occupy smaller, not larger fraction of the Sahara and Arabian deserts, but this sentence is indeed not very clear. In the revised manuscript, we will improve it in the following way:

"... They are, for instance, excluded from the dry inner part of Australia, and they also occupy smaller areas compared to light and dark cyanobacteria-dominated biocrusts in the Sahara and the Arabian desert."

14. From Figure 4e, I would conclude the light cyanobacteria are the outlier under low

Fig. A8 NEW Total biocrust surface cover for (a) doubled dew availability (b) increased surface resistance to evaporation (from 0 to  $100 \text{ sm}^{-1}$ ).

precipitation, rather than lumping the dark cyanobacteria together with them, which is done throughout the text.

We assume that the referee actually refers to Fig. 4f, which shows the dependence of relative biocrust cover on annual rainfall, since Fig. 4e only shows total biocrust cover. We are not sure what is meant by "lumping together", but we assume that this refers to discussing patterns of light and dark cyanobacteria together. We use the term "light and dark cyanobacteria-dominated biocrusts" only at two locations in the manuscript (second paragraph of Results section, P 11 and beginning of section 4.1). In both cases, we refer to the different latitudinal patterns of light and dark cyanobacteria-dominated biocrusts compared to chlorolichen and moss-dominated crusts, where the former are more abundant at low latitudes and the latter are more abundant at higher latitudes.

We agree with the referee that it is misleading to state that light and dark cyanobacteriadominated biocrusts occur in dry regions and chlorolichen and moss-dominated crusts in wet regions (section 4.1), since Fig. 4f indeed shows a more differentiated picture of the biocrust type - rainfall relation. Moreover, the relative cover of the different biocrust types also shows a dependence on temperature, which we did not discuss in the original manuscript. To provide a more detailed and understandable description of the climatic factors which influence relative cover of crust types in the model, we will change the respective parts of the Results and Discussion sections in the revised manuscript, and we will replace Fig. 4 (f) by a new figure, which shows the dependence of relative cover both on temperature and rainfall. In the Results section, we will state the following (P 11 L 4):

"In general, moss- and chlorolichen-dominated biocrusts are more abundant in more poleward regions and light and dark cyanobacteria-dominated biocrusts are more abundant in regions at low latitudes."

and (P 11 L 9):

"In Fig. (5,NEW) we show the dependence of the simulated relative cover of the different biocrust types on the amount of rainfall and the average temperature. In general, the relative cover fractions of light and dark cyanobacteria-dominated biocrusts

increase with warmer and drier climatic conditions, while the share of chlorolichen- and moss-dominated biocrusts on the total coverage increases for cooler and wetter conditions. For regions with the lowest rainfall, only light cyanobacteria-dominated biocrusts occur in the model. If rainfall slightly increases, the relative cover of dark cyanobacteria-dominated biocrusts rises, until they are equally abundant as the light cyanobacteria-dominated crusts. The slope of this increase, however, depends on temperature (Fig. 5 (a,b)): Dark cyanobacteria-dominated biocrusts increase in abundance faster under cooler climatic conditions. The response of the relative cover fractions of chlorolichen-and moss-dominated biocrusts to temperature and rainfall is similar: They are both absent under the warmest and driest climatic conditions, but under cool and wet conditions they have the largest share on the total cover. Chlorolichen-dominated crusts seem to be grow slightly better under cool and dry conditions compared to moss-dominated crusts (Fig. 5 (a,c)). The dependence of biocrust type on amount of rainfall and average temperature is also reflected in the average relative cover values per biome simulated by LiBry (see Tab. 1)."

Furthermore, we will change the first two paragraphs of section 4.1 to better explain the relation between simulated patterns of biocrust types and the processes implemented in the model (see also comment 16 below):

"We find a dependence of biocrust type on the amount of rainfall and average temperature (Fig. 5 NEW). In the driest regions, light cyanobacteria-dominated biocrusts are most abundant. With increasing rainfall, the relative cover of dark cyanobacteriadominated crusts increases, followed by chlorolichen- and moss-dominated biocrusts. With decreasing temperature, chlorolichen- and moss-dominated biocrusts become increasingly abundant. At cool and wet conditions, they are more abundant than light and dark cyanobacteria-dominated biocrusts, which has also been reported from field studies [Bowker et al., 2006, Kidron et al., 2010, Büdel et al., 2009].

To understand why the LiBry model predicts these climate-driven spatial patterns of different biocrusts types, several physiological processes have to be considered, which are implemented in the model. First, the dynamic cover scheme requires all strategies to compensate losses in their surface cover through growth, in order to survive. These losses result from disturbance and turnover of biomass and they are not directly related to climate. Growth, however, is proportional to the difference between photosynthesis and respiration, which means that it decreases with drier conditions, which reduce the amount of active time, and it also decreases with higher temperatures, since respiration increases with temperature in the model. Consequently, markedly dry and warm conditions are unfavorable in general for the simulated strategies. Secondly, smaller strategies have the advantage of a more efficient cover expansion in the model, since they can produce more surface area for a given amount of biomass growth than taller strategies. This means that they are more likely to maintain their cover against disturbance and turnover under unfavorable climatic conditions. Since we defined small strategies (less than 2 mm) as cyanobacteria, this explains the high relative cover of light- and dark cyanobacteria-dominated biocrusts under warm and dry conditions. Fig. 5 shows that dark cyanobacteria-dominated crusts do not perform as well as light cyanobacteria-dominated crusts under the warmest and driest conditions. This

---

## Author Comment (AC2) · 9 Feb 2019

**Global NO and HONO emissions of biological soil crusts estimated by a process-based non-vascular vegetation model**

Philipp Porada[1,2], Alexandra Tamm[2], Jose Raggio[3], Axel Kleidon[4], Ulrich Pöschl[2], and Bettina Weber[2]

[1]University of Potsdam, Vegetation Ecology and Nature Conservation, Am Mühlenberg 3, 14476 Potsdam, Germany
[2]Max Planck Institute for Chemistry, Multiphase Chemistry Department, P.O. Box 3060, 55020 Mainz, Germany
[3]Departamento de Biología Vegetal II, Facultad de Farmacia, Universidad Complutense de Madrid, Madrid, Spain
[4]Max Planck Institute for Biogeochemistry, P.O. Box 10 01 64, 07701 Jena, Germany

We thank the referees for their useful and thorough comments which will be helpful to improve our manuscript. We have prepared a detailed response to the points raised by the referees. We show the referees' comments in italic text, while our responses are formatted as standard text.

**Response to the comments of referee #1**

*General comments: This study provided an estimate of global NO and HONO emissions from biocrusts by improving a non-vascular vegetation model (LiBry). Specifically, the authors improved the model to predict relative cover of different biocrust types and then estimated NO and HONO emissions through extrapolating the relationship between NO and HONO emissions and water content (i.e. Figure 1). While the authors did some interesting works, I think there have a couple of major deficiencies in this study. First, the authors did not justify the worldwide applicability of the relationship between NO and HONO emissions and water content while this relationship is critical for the products (global NO and HONO emissions) presented in this manuscript. Second, the authors did very limited model validation for the NO and HONO emissions. These deficiencies make me feel the major products, global NO and HONO emissions, are not very solid in the current version of the manuscript. Therefore, I think substantial works/improvements are necessary for addressing the deficiencies I mentioned.*

We are glad that the referee thinks that our study is interesting. We agree that the uncertainty associated with our estimate on NO and HONO emissions could be better

analysed and constrained. In the revised manuscript, we will point out more clearly the uncertainty of our estimate. We will extend the abstract (before the last sentence):

"Based on this, we estimate global total values of $1.04\,\mathrm{Tg\,yr^{-1}}$ NO-N and $0.69\,\mathrm{Tg\,yr^{-1}}$ HONO-N released by biological soil crusts. Due to the low number of observations on NO and HONO emissions suitable to validate the model, our estimates are still relatively uncertain. However, they are consistent with the amount estimated by the empirical approach, which confirms that biological soil crusts are likely to have a strong impact on global atmospheric chemistry via emissions of NO and HONO."

Moreover, we found one additional study which reports NO emissions by biocrusts, which we will add to the revised manuscript (P 15 L 23):

"Barger et al. [2005] estimate annual NO emissions of 2 to 16 mg m$^{-2}$ yr$^{-1}$ by biological soil crusts at two fields sites in the Canyonlands National Park, Utah, USA. This compares well to the large-scale estimate of 10 to 20 mg m$^{-2}$ yr$^{-1}$ of NO-N simulated by LiBry for this region."

In the revised discussion, we will add (P 23 L 2):

"Due to the paucity of studies which report observation-based estimates of NO and HONO emissions by biological soil crusts, the validation of our model is relatively limited in this regard. To obtain annual values for NO and HONO emissions, usually short-term measurements on wetted biological soil crust samples are extrapolated to the whole year based on the number of wetting events at the respective field site. Hence, for a future, more detailed validation of the LiBry model it would be useful to create data sets which include both climate data with high temporal resolution and also NO and HONO emissions from the same field site. In addition, activity of the biocrusts should be monitored, so potential mismatches between model and observations can be traced to either the simulation of the dynamic water content or the specific emissions per crust type."

To better quantify the uncertainty of our estimate, we carried out additional sensitivity analyses on the relationship between water saturation of biological soil crusts and their NO and HONO emissions. These will be described as follows in the Methods section of the revised manuscript (P 10 L 20):

"Moreover, we account for uncertainty resulting from variation in the measured relations between water saturation and NO and HONO emissions of different biocrust types.

(f) We calculated the standard deviation of the measurements made by [Weber et al., 2015], and subtracted it from the average curves shown in Fig. 1 to create a lower bound of NO and HONO emissions as a function of biocrust water content. To create a corresponding upper bound, we added one standard deviation to the curves.

(g) We replaced our default relationship between water content and NO and HONO emissions of different biocrust types by an alternative one etablished by Meusel et al. [2018]. They used a similar approach, but for a different location, a field site in Cyprus."

In the Results section, we will add the following (P 18 L 14, see also extended Tab. 2 below at point 12 of referee #2):

"Simulated NO and HONO emissions are also sensitive to variation in the relationship between water content and emissions. Decreasing the specific emissions at a given water

content by one standard deviation reduces simulated total global NO emissions by 25% and HONO emissions by 50% compared to the originally estimated emissions (see Tab. 2). Increasing specific emissions by one standard deviation raises simulated NO emissions by 85% and HONO emissions by 107%. Moreover, replacing our default relation between water content and NO and HONO emissions, which is based on Weber et al. [2015], by an alternative relation derived from Meusel et al. [2018], significantly affects our estimates. While simulated NO emissions decrease by 66%, HONO emissions remain almost unchanged and are reduced by only 3%."

We will also update the Discussion with the following text (P 22 L 31):

"The sensitivity analysis demonstrates that varying the relation between water saturation and NO and HONO emissions of biocrusts by one standard deviation leads to relatively large changes in the estimated NO and HONO emissions. Also, replacing the default relation by an alternative one which is based on measurements from a different field site significantly affects simulated NO and HONO emissions by biocrusts. Further analyses of the dependence of NO and HONO emissions on the water saturation state of biocrusts are needed to determine the causes of this large variation. A potential first step in this direction would be to determine the dynamic nitrogen content of biocrusts and also of the underlying soil. Subsequently, the nitrogen content could be related to variation in the relationship between NO / HONO emissions and biocrust type. It is likely that emissions will increase with soil nitrogen content, which means that biocrusts will contribute relatively more to total emissions in areas with nitrogen-poor soils, and less on nitrogen-rich soils. Including this factor in our modeling approach would lead to a more differentiated global pattern of NO and HONO emissions. Furthermore, processes which control the non-linear dependence of NO and HONO emissions on water saturation need to be clarified. One possible explanation for the decrease of emissions at high water saturation is the limitation of the microbes which produce NO and HONO by increasingly low oxygen supply. Differences in the structure between biocrust types and, consequently, their diffusivity for oxygen, may then explain variation in the shape of the relation between water saturation and NO / HONO emissions. A fully process-based scheme of NO and HONO emissions by biocrusts may contribute to further quantifying nitrogen cycling in dryland soils. Currently, it is still difficult to close the nitrogen balance for these areas due to uncertainty regarding the amount of both nitrogen inputs via atmospheric deposition and biotic fixation and also outputs through various gaseous losses, leaching and erosion."

*Other specific points: Page 2 Line 19: Grammar error.*

In the revised version, we will change this sentence (see point 2 by referee #2).

*Page 2 Lines 33 to 34: Grammar error in this sentence.*

We are not able to find a grammar error in this sentence, but this issue may be resolved during the copy-editing.

*Page 4 Line 29: Add a full stop.*

We will add a full stop in the revised manuscript.

*Page 5 Line 15: How reliable of this relationship/method? How about applicability and uncertainties of this relationship/method?*

In the revised manuscript, we will discuss in more detail the potential reasons for variation in the relationship between water content and NO and HONO emissions of biocrusts. Moreover, we will quantify the sensitivity of the model estimates to variation in this relationship (see our first point above).

*Page 9 Line 26: Past tense should be used for what you did. Please make necessary changes throughout the MS.*

In the revised version of the manuscript, we will use past tense for the description of our work.

*Page 11 Lines 26 to 27: This sentence is not clear.*

We agree with the referee that this sentence can be improved and we will change it in the revised manuscript as follows:

"The large-scale spatial pattern of active time shows highest values in more poleward regions, medium values in large parts of (sub)tropical regions and lowest values in desert regions. This may be explained by a combination of the patterns of rainfall and surface temperature (see Fig. A6): In poleward regions, rainfall is relatively high and evaporation is moderate, due to lower surface temperatures, resulting in relatively large water supply. In many tropical regions, rainfall is also high but evaporation is markedly increased compared to more poleward regions, as indicated by the higher surface temperatures. This results in less available water and thus in less active time."

*Page 15 Line 1: The sub-titles of the result section are confusing. You described a lot of results before 3.1 Global validation; so what is the sub-title for the results described before 3.1?*

We will add another subtitle called "Global patterns of biocrust cover and NO and HONO emissions" for the beginning of the Results section.

*Page 15 Line 15: report should be reported. This issue has been found throughout the manuscript.*

We will change the revised manuscript in this regard.

*Page 18 Line 14: Is the low sensitivity due to the simple model for NO and HONO emissions?*

The main reason is the low sensitivity of active time to the parameters tested in the original manuscript (see Tab. 2). In the revised manuscript, we will include an extended sensitivity analysis, which also focuses on the water balance of the organisms. Here, we partly find a higher sensitivity of NO and HONO emissions, as discussed in the response to the comments of referee #2 (point 12).

*Page 20 Line 30 to 33: It might be worthy to provide some explanations for this statement. Why this method is more appropriate than calibration?*

We agree that this should be better explained. In the revised manuscript, we will provide a more detailed reasoning, where we argue that the physiological properties which are used to distinguish between the different biocrust types are not the only uncertain parameters in our approach. Hence, by calibrating the model with regard to these physiological properties, we would implicitly assume that all other uncertain parameters have the correct values, which is not necessarily true (see also point 17 by referee #2).

*Page 21 Line 19: that should be than.*

We will correct this in the revised manuscript.

*Page 22 Line 30 to 33: Does NO and HONO emissions affected by other factors in bicrust systems, such as N content etc?*

In reality, the N contents of the biocrust and also the underlying soil most likely influence the shape and magnitude of the relationship between water saturation and NO / HONO emissions. In the revised manuscript, we will extend the discussion in this regard (see our first point above):

**Response to the comments of referee #2**

*The Porada et al. article uses a process-based modeling approach with the LiBry model to determine the global areal coverage of biocrust types responsible for emitting NO and HONO, which are important atmospheric chemical constituents for OH. They then relate biocrust type to water saturation to determine emissions of these two constituents. The paper is well written, has sufficient validation and sensitivity analysis, and concludes with values similar to an empirical upscaling study, but for a more restrained area, so that, in fact, the emissions are actually even larger. The wetting event validation comes from four sites in a single South African location, so would be good if there was an attempt for more global validation of this variable, as it is key to how emission are modeled. There are quite a few things that can be revised to help clarify the paper, as suggested below.*

We are glad that the referee makes a positive assessment on our paper, in particular regarding validation and sensitivity analysis. We agree with the referee that the manuscript would benefit from a more extended validation of the simulated dynamic water saturation. Therefore, in the revised version of our manuscript, we compare the model to observational data from another site with a different climate, which is located in South Spain, and we also carry out an extended sensitivity analysis with respect to the dynamic water content (see responses to comments 12 and 20 below for details).

*1. The four biocrust types are light and dark cyanobacteria, chlorolichen, and mosses, yet the LiBry model is Lichen and Bryophyte. At times bryophyte is used instead of mosses (p. 20 line 8), so should just specify that up front and then use mosses consistently throughout.*

We agree that this can be made clearer, and we will explain in the revised manuscript at the beginning of Sect. 2 that we use "mosses" here instead of "bryophytes".

*2. P. 2, line 19  100% of N2O emissions in dryland regions is what percent of total N2O emissions?*

We will change this sentence in the revised manuscript to the following:
  "The authors found that these organisms are responsible for $4$-$9\%$ of global emissions of $N_2O$ from natural terrestrial sources. In drylands, where they represent main components of biocrusts, they may even contribute up to 100% to $N_2O$ emissions."

*3. More explanation of physiological strategies in LiBry. A few of them are used to partition between the four types as in Figure 2. But in section 2.4 (first sentence) it is mentioned that there 3000 physiological strategies, so it is not clear what all these strategies are referring to. Also, some further explanation of how photosynthetic capacity, height, and CO 2 diffusivity are used in the model would be helpful.*

We agree with the referee that the description of the physiological strategies and their relation to different biocrust types could be more detailed. In fact, not only a few of the strategies are used to partition between the biocrust types, but all of them, which means that each of the 3000 strategies is assigned to one of the four crust types at the start of the simulation. In the revised manuscript, we will therefore extend the corresponding paragraph (P 6 L 3-10) by the following text:

  "To assign strategies to biocrust types, we first determine to which group of photoautotrophs (lichens, mosses or cyanobacteria) each simulated physiological strategy belongs. This is necessary since the LiBry model does not categorize the simulated strategies by default. Instead, an individual strategy is defined only through its unique combination of values of several physiological parameters, as described above. We use these physiological parameters to distinguish the strategies into lichens, mosses and cyanobacteria. For this purpose, the following parameters are taken into account: Height, $CO_2$-

diffusivity in the wet state, and photosynthetic capacity. The growth height of a strategy has several effects in the model: For the same amount of cover expansion, the higher a strategy is, the more biomass is needed, which is a competitve disadvantage. However, taller strategies have more potential to store water per given area, and they may also outcompete smaller strategies with regard to light availability. The $CO_2$-diffusivity at high water saturation is an important physiological constraint, since organisms with higher diffusivity are able to grow more than those with low diffusivity in the model. This advantage is, however, associated with increased loss of water through evaporation for given climatic conditions, due to the more open structure of the biomass. Photosynthetic capacity controls the ability of a photoautotroph to use high light intensities and to capture $CO_2$ from the atmosphere. Strategies with a high photosynthetic capacity are able to grow more than those with low capacity under certain climatic conditions, but this advantage comes at the cost of increased maintenance respiration and turnover. We want to mention that the categorization of strategies into lichens, mosses and cyanobacteria has no impact on the dynamics of the vegetation in the model, it only affects the simulated NO and HONO emissions."

*4. At end of Introduction there are two main extensions of LiBry mentioned - assigning physiological strategies to the four types (so, is this really just defining new types by physiological strategies - would make it clearer to describe it in this way); the dynamic surface cover model described in Figure 3; and how about adding determining emissions of NO and HONO to the model?*

We agree that these points should be clarified and we will change the end of the Introduction (P 3 L 24) accordingly:

".. To this end, we extend the LiBry model in three central aspects: First, we introduce a scheme which categorizes the large number of physiological strategies simulated by LiBry for drylands into lichens, mosses and cyanobacteria. We then define the different biocrust types considered in the study by Weber et al. [2015] according to these vegetation groups. This enables us to take into account the strong differences in NO and HONO emissions between biocrust types. Secondly, we alter the scheme for dynamic surface cover of the physiological strategies in LiBry, which enables us to predict the relative cover of each biocrust type. Thirdly, we extend LiBry by an empirical scheme which calculates NO and HONO emissions of different biocrust types based on their water saturation. Thereby, saturation of the biocrusts is based on the dynamic water content of the individual physiological strategies simulated by LiBry. We evaluate our estimates of biocrust surface cover both at the local and the global scale by comparison to observations and we compare simulated NO and HONO emissions to the available estimates from the literature."

*5. The fundamental assumptions from Figure 1 that underlie the emissions model are pretty simplistic. So, how about further discussion of what controls water saturation - not just in terms of water balance, but how important is the uptake of water by the photosynthetic biocrusts?*

We agree with the referee that it is useful to describe water relations in more detail and we will therefore extend the model description by the following text (P 4 L 16):

"Several physiological properties regulate the dynamic water saturation of non-vascular vegetation in the model: First, the uptake of water is limited not only by rainfall or snow melt, but also by the water storage capacity of the organisms, which depends on it height and the porosity of the biomass. At full saturation, additional water input infiltrates into the soil. The extent to which dew can be used as a water source in the model depends mostly on climatic conditions, and to a limited extent on properties of the organisms which influence the surface temperature, e.g. via evaporation. Secondly, also water loss is regulated by properties of the organisms. These are the same as for water uptake, namely the specific water storage capacity, capillary structure of the biomass and albedo. Note that non-vascular vegetation does not possess stomata, so an active reduction of evaporation is not possible."

We will also add more explanations on how water saturation is related to NO and HONO emissions (see below).

*6. NO and HONO emission are based on water saturation and Q10 - is it simply these two terms multiplied? Include an equation that explicitly states how it is calculated.*

To derive NO and HONO emissions from water saturation, we created a look-up table based on a discetization of the measured curves shown in Fig.1. The values of NO and HONO emissions for a given value of water saturation were subsequently scaled as a function of temperature based on a $Q_{10}$-relationship. Therefore, we cannot provide an equation for the complete calculation of the emissions. Instead, we will explain this approach in more detail and include the following text in the revised manuscript (P 5 L 15):

"To implement these relations into the LiBry model, we discretize the curves shown in Fig. 1 and create a look-up table, which assigns values of NO and HONO emissions for each value of water saturation. Subsequently, the emissions are scaled according to surface temperature:

$$E_{\mathrm{NO,ONO}} = E_{\mathrm{NO,HONO}}(\Theta)Q^{\frac{T_\mathrm{S}-T_\mathrm{REF}}{10.0}} \tag{1}$$

where $E_{\mathrm{NO,HONO}}$ are the emissions of NO and HONO, respectively, $E_{\mathrm{NO,HONO}}(\Theta)$ are the emissions at a given water saturation $\Theta$ based on the look-up table, $Q$ is the $Q_{10}$-value, and $T_\mathrm{S} - T_\mathrm{REF}$ is the difference between the surface temperature of the simulated organisms and the reference temperature. In this way, NO and HONO emissions are calculated from the simulated water saturation at each time step of the model run."

*7. Since the effect of nitrogen cycling is not included (p. 5, lines 16-17), there should be some discussion of that in the Discussion as to how that may change things.*

We agree that the Discussion should be extended in this regard and we will add a corresponding text to the revised manuscript (see first point of the response to referee #1).

*8. Figure 3: I suggest changing the green colors so they grade from darker to lighter with height, rather than the lightest one in the middle.*

We will change this accordingly in the revised manuscript.

*9. Need reference for the WATCH data (p. 8, line 13)*

We cite Weedon et al. [2011] in the manuscript as this is the recommended reference. To make this more clear, we will extend the reference in the following way:

"Climate forcing data are based on the WATCH data set (see Weedon et al. [2011] and also http://www.eu-watch.org/data_availability), which span the years 1958 to 2001."

*10. I would like to see more explanation of how disturbance is applied (p. 9, line 6) - although the Porada reference is given, one or two sentences here would be helpful.*

We will add in the revised manuscript the following, to explain this in more detail (P 9 L 7):

".. This means, that for each of the 16 biomes which are considered in LiBry at the global scale, a characteristic interval between disturbance events is determined from the literature. Thereby, we account for processes such as fire, windbreak of trees, which destroys the habitat of epiphytic non-vascular vegetation, and also trampling by animals. The disturbance interval is then converted into the fraction of the biocrust surface cover which is destroyed once per month in the simulation, e.g an interval of 8 years would result in roughly 1 % of the surface cover being disturbed each month."

*11. Table A2: list units as %.*

We will add the unit % to the table header.

*12. Sensitivity to model parameters (end of Methods) is based on those that affect total biocrust cover, relative cover, temperature-dependence of NO and HONO emissions, but what about those that affect water saturation? There ought to be some test of a key parameter here, as that affects NO and HONO emissions.*

We agree with the referee that it would be useful to extend the sensitivity analysis in this regard. Therefore, we have carried out additional tests, which we will describe in the revised manuscript in the following way (end of Methods):

[revised manuscript text omitted]

Finally, we will discuss the outcomes of the sensitivity analysis in the Discussion (P 21 L 26):

"The relatively large sensitivity of our estimated NO and HONO emissions to parameters which influence the dynamic water content is plausible. If maximum available dew is doubled, or a resistance to evaporation is introduced, the potential area where biocrusts may occur in the model is significantly larger and total biocrust cover and active time significantly increase (Tab. 2). The strong increase in NO and HONO emissions can further be explained by the large increase in biocrust cover in warm regions, where high temperatures further enhance NO and HONO release (Eq. 1). However, the simulated global patterns of biocrust coverage under doubled dew or increased surface resistance seem to be inconsistent with observations. In extremely dry regions of North Africa, Arabia and Australia, high cover values of more than 70% are simulated, which are substantially higher than large-scale average values which are commonly assumed for these regions (see Fig. A8 NEW). Moreover, although dew of 80 mm yr$^{-1}$ may occur locally under certain conditions in drylands, this value is most likely too high for a large-scale estimate (see also Vuollekoski et al. [2015] for typical values),"

*13. P. 11, line 2 - why are chlorolichen-dominated crusts larger fractions of the Sahara and Arabian deserts - does not look that way from the figure?*

We actually wanted to say that chlorolichen-dominated crusts occupy smaller, not larger fraction of the Sahara and Arabian deserts, but this sentence is indeed not very clear. In the revised manuscript, we will improve it in the following way:

".. They are, for instance, excluded from the dry inner part of Australia, and they also occupy smaller areas compared to light and dark cyanobacteria-dominated biocrusts in the Sahara and the Arabian desert."

*14. From Figure 4e, I would conclude the light cyanobacteria are the outlier under low*

[Figure]

**Fig. A8 NEW** Total biocrust surface cover for (a) doubled dew availability (b) increased surface resistance to evaporation (from 0 to 100 s m$^{-1}$).

*precipitation, rather than lumping the dark cyanobacteria together with them, which is done throughout the text.*

We assume that the referee actually refers to Fig. 4f, which shows the dependence of relative biocrust cover on annual rainfall, since Fig. 4e only shows total biocrust cover. We are not sure what is meant by "lumping together", but we assume that this refers to discussing patterns of light and dark cyanobacteria together. We use the term "light and dark cyanobacteria-dominated biocrusts" only at two locations in the manuscript (second paragraph of Results section, P 11 and beginning of section 4.1). In both cases, we refer to the different latitudinal patterns of light and dark cyanobacteria-dominated biocrusts compared to chlorolichen and moss-dominated crusts, where the former are more abundant at low latitudes and the latter are more abundant at higher latitudes.

We agree with the referee that it is misleading to state that light and dark cyanobacteria-dominated biocrusts occur in dry regions and chlorolichen and moss-dominated crusts in wet regions (section 4.1), since Fig. 4f indeed shows a more differentiated picture of the biocrust type - rainfall relation. Moreover, the relative cover of the different biocrust types also shows a dependence on temperature, which we did not discuss in the original manuscript. To provide a more detailed and understandable description of the climatic factors which influence relative cover of crust types in the model, we will change the respective parts of the Results and Discussion sections in the revised manuscript, and we will replace Fig. 4 (f) by a new figure, which shows the dependence of relative cover both on temperature and rainfall. In the Results section, we will state the following (P 11 L 4):

"In general, moss- and chlorolichen-dominated biocrusts are more abundant in more poleward regions and light and dark cyanobacteria-dominated biocrusts are more abundant in regions at low latitudes."

and (P 11 L 9):

"In Fig. (5,NEW) we show the dependence of the simulated relative cover of the different biocrust types on the amount of rainfall and the average temperature. In general, the relative cover fractions of light and dark cyanobacteria-dominated biocrusts

increase with warmer and drier climatic conditions, while the share of chlorolichen- and moss-dominated biocrusts on the total coverage increases for cooler and wetter conditions. For regions with the lowest rainfall, only light cyanobacteria-dominated biocrusts occur in the model. If rainfall slightly increases, the relative cover of dark cyanobacteria-dominated biocrusts rises, until they are equally abundant as the light cyanobacteria-dominated crusts. The slope of this increase, however, depends on temperature (Fig. 5 (a,b)): Dark cyanobacteria-dominated biocrusts increase in abundance faster under cooler climatic conditions. The response of the relative cover fractions of chlorolichen- and moss-dominated biocrusts to temperature and rainfall is similar: They are both absent under the warmest and driest climatic conditions, but under cool and wet conditions they have the largest share on the total cover. Chlorolichen-dominated crusts seem to be grow slightly better under cool and dry conditions compared to moss-dominated crusts (Fig. 5 (a,c)). The dependence of biocrust type on amount of rainfall and average temperature is also reflected in the average relative cover values per biome simulated by LiBry (see Tab. 1)."

Furthermore, we will change the first two paragraphs of section 4.1 to better explain the relation between simulated patterns of biocrust types and the processes implemented in the model (see also comment 16 below):

"We find a dependence of biocrust type on the amount of rainfall and average temperature (Fig. 5 NEW). In the driest regions, light cyanobacteria-dominated biocrusts are most abundant. With increasing rainfall, the relative cover of dark cyanobacteria-dominated crusts increases, followed by chlorolichen- and moss-dominated biocrusts. With decreasing temperature, chlorolichen- and moss-dominated biocrusts become increasingly abundant. At cool and wet conditions, they are more abundant than light and dark cyanobacteria-dominated biocrusts, which has also been reported from field studies [Bowker et al., 2006, Kidron et al., 2010, Büdel et al., 2009].

To understand why the LiBry model predicts these climate-driven spatial patterns of different biocrusts types, several physiological processes have to be considered, which are implemented in the model. First, the dynamic cover scheme requires all strategies to compensate losses in their surface cover through growth, in order to survive. These losses result from disturbance and turnover of biomass and they are not directly related to climate. Growth, however, is proportional to the difference between photosynthesis and respiration, which means that it decreases with drier conditions, which reduce the amount of active time, and it also decreases with higher temperatures, since respiration increases with temperature in the model. Consequently, markedly dry and warm conditions are unfavorable in general for the simulated strategies. Secondly, smaller strategies have the advantage of a more efficient cover expansion in the model, since they can produce more surface area for a given amount of biomass growth than taller strategies. This means that they are more likely to maintain their cover against disturbance and turnover under unfavorable climatic conditions. Since we defined small strategies (less than 2 mm) as cyanobacteria, this explains the high relative cover of light- and dark cyanobacteria-dominated biocrusts under warm and dry conditions. Fig. 5 shows that dark cyanobacteria-dominated crusts do not perform as well as light cyanobacteria-dominated crusts under the warmest and driest conditions. This

[Figure]

**Fig. 5 NEW** Dependence of relative cover of different biocrust types on annual rainfall and average temperature. (a - d) Only a subset of all grid cells in the study area are included in the relations, depending on the climatic conditions specified in the upper left corner of the plots.

can be explained by another physiological trade-off implemented in the model, which links photosynthetic capacity to maintenance respiration. We have defined that small strategies with a high photosynthetic capacity and thus a high respiration rate belong to dark cyanobacteria-dominated crusts, while those with low photosynthetic capacity and respiration are assigned to light cyanobacteria-dominated crusts (Fig. 2). Since respiration increases stronger than photosynthesis at high temperatures, strategies with a high 'baseline' respiration may grow less under warm conditions, which explains why dark cyanobacteria-dominated crusts are less abundant in the warmest regions. Moreover dark cyanobacteria-dominated crusts are more negatively affected by reduced active time due to dry climate compared to light cyanobacteria-dominated biocrusts, also at moderate temperatures (Fig. 5(c) NEW). A potential reason for this is the increase in turnover rate with higher photosynthetic capacity, which is not reduced as strongly as growth during periods with low activity. This means that dark cyanobacteria-dominated

crusts can use their potential for stronger growth due to their higher photosynthetic capacity only if sufficient active time is available (and sufficient radiation). For this reason, their relative cover is maximal under moderate temperatures and sufficiently wet climatic conditions (Fig. 5(d) NEW). Finally, the growth height of a simulated strategy represents a competitive advantage in the model. If two strategies compete for the same location of available free area (Fig. 3), the taller strategy will be able to overgrow the smaller one. Hence, under favourable climatic conditions (moderate temperatures and sufficient rainfall), simulated lichens and bryophytes may partly outgrow cyanobacteria, and therefore chlorolichen- and moss-dominated biocrusts have the largest share on the total cover (Fig. 5 NEW). The higher relative cover of chlorolichen-dominated biocrusts compared to moss-dominated crusts at cooler and drier conditions (Fig. 5 (a,c) NEW) may be explained by another physiological trade-off in the LiBry model: Strategies with a high diffusivity for $CO_2$ during the saturated state can grow more under favourable conditions than strategies with a low $CO_2$-diffusivity. However, due to their more open structure, they also evaporate more water, and thus become limited faster under dry conditions and are less productive. Since we have defined in the model that mosses have a higher $CO_2$-diffusivity and higher evaporation than lichens on average, the simulated moss-dominated biocrusts may be more affected by dry climate and are thus less abundant than chlorolichen-dominated crusts. This pattern is more pronounced for cool climatic conditions compared to warm conditions (Fig 5 (a,b) NEW). A possible reason for this is that the higher resistance of lichens against evaporation is not sufficient to prevent desiccation under warm climatic conditions, which means that they have no advantage over mosses anymore. Large water compensation and saturation values of bryophytes have also been experienced during $CO_2$ gas exchange measurements under controlled conditions [Tamm et al., 2018, Raggio et al., 2018]."

*15. Appendix figures are out of order - A6 is discussed first, then A2-A5, then A1, and finally A7. Please put these in order. Also, from Figure 9 and Figure A2-A5 I am unclear which site (1 - 4) is being referred to, so there should be some way of distinguishing between the four sites. Furthermore, why are the results from only one of these sites shown in the main text?*

We will rearrange the sequence of figures in the appendix, as requested by the referee. We will also indicate in Fig. 9 which of the sites from Figs. A3 and A5 are shown. The main reason why we show only one site and one biocrust type in the main text is that the differences between the sites are not very large, and also the crust types differ substantially only in the amount of moisture content, but not in their patterns of activity. To save space in the main text, we therefore only show one example and provide the rest of the figures in the appendix.

*16. In section 4.1, there should be better attribution of these explanations as to what actually occurs in the model, to distinguish from general arguments. For example, why is there a competitive advantage due to height given that shading is not an issue? How does the discussion of moss vs lichen in dry/wet conditions pertain to the model design?*

*Also, p. 20 line 4 - not sure how this follows previous sentence.*

We agree with the referee that the links between simulated patterns of relative cover of biocrusts types and the process implemented in the model should be better explained. We have therefore provided an updated section 4.1, described under point 14 above. The advantage of taller strategies over smaller ones in the model is due to their ability to overgrow them, which means that a layer of cyanobacteria can be overgrown by a lichen or moss, for instance, which we think is realistic. We have described above (point 14) in more detail how dry/wet conditions influence relative cover of mosses and lichens in the model. We also have changed the sentence on P 20, L 4 of the original manuscript to clarify this, as described in point 14 above.

*17. P. 20, line 34 - it is ok for process-based models to determine parameters based on optimization from field results - do not need to just use values based on the literature.*

We agree that process-based model can be calibrated against field data in general. To make our point more clear, we will change this sentence to the following in the revised manuscript:

"However, the relative cover of the different biocrusts types simulated by LiBry does not only depend on the two physiological properties, $CO_2$-diffusivity and photosynthetic capacity, which were used to distinguish the crust types, but also on other factors, such as disturbance interval, for instance. By calibrating the model with regard to $CO_2$-diffusivity and photosynthetic capacity, we would implicitly assume that all other uncertain parameters have the correct values, which is not necessarily true."

*18. P. 21, line 5: Are the results not sensitive to climate or is it just that at this one site the match is good?*

As we have shown in e.g. Fig 4 and 5 (NEW), the model estimates are markedly sensitive to climate. What we wanted to test by using the global data for the location in South Africa is (a) if there are large differences in the climate data and (b) how these differences influence the model estimates. We found no large differences between global and local data and, consequently, the estimated cover values were similar. This is not the case, however, for the additional site in Almería. We will formulate this more precisely in the revised manuscript:

"We find that the global climate data for the grid cell which includes Soebatsfontein are a good approximation to the climate data from the local station (Fig. A1 (*will be renumbered*)). Consequently, the estimated values of cover of biocrusts types based on the different climate input data are similar. However, for the site near Almería, the local data represent significantly drier conditions than the large-scale data for South Spain, which explains the higher activity simulated by LiBry for Spain in general compared to the Almería site. This illustrates that the global data are not necessarily a good approximation for those local conditions which stronlgy deviate from the regional climate."

*19. P. 21, line 23: Would changing the albedo throw off the other seasons, or is it a change that only occurs in the warm season?*

To change the albedo, we would have to prescribe this parameter for the simulated strategies, e.g. by narrowing down the range of albedo values which are possible for the strategies at the start of the simulation. Probably, less strategies would survive under a prescribed lower albedo due to higher surface temperature, and this would affect our estimates. To change this, we would have to implement damage by UV radiation in the model, so that strategies with lower albedo would not only face the disadvantage of higher surface temperature, but also benefit from UV protection.

*20. P. 22, line 23: The conclusion of correct wetting events is from only the one location in South Africa, at least as presented in this paper. There really needs to be a more broadscale analysis globally of this in order to make this conclusion. Is that possible to do - as that is really key to getting the emissions correct?*

We agree with the referee that a more extended validation of the dynamic water content of biocrusts would be useful. However, it is difficult to find data sets which include both local climate data and also observations of wetting events or metabolic activity at high temporal resolution. We were able to get access to one additional suitable set of observational data from a site close to Almería in South Spain. We find that the LiBry model reproduces well the annual pattern of metabolic activity (see Fig. 9 NEW), which supports our large-scale estimate of active time of biocrusts and the associated NO and HONO emissions. In the revised manuscript, we will change the following (Methods section 2.5, P 9 L 34):

"We also assess the model performance for another local site, located close to Almería in South Spain [J. et al., 2017], where metabolic activity of biocrusts was observed, which is closely connected to their dynamic water content. Consequently, we compare simulated active time to the field observations. It should be mentioned that not all input variables needed for running the LiBry model were available from the two study sites. While the most important variables solar radiation, rainfall, air temperature, and relative humidity were available, we used atmospheric downwelling longwave radiation from our global data set and, for Soebatsfontein, also wind speed. Moreover, we used leaf and stem area index from the global data, which are needed to compute shading effects, but are very low for the regions of Soebatsfontein and Almería and thus hardly affect the results. Due to lack of knowledge on the disturbance regime, we assume the standard disturbance interval of 100 years used in LiBry for the desert biome. All data from Soebatsfontein were obtained between October 2008 and October 2009, and the data from Almería are from 2013. The generally low availability of data sets which include biocrust cover together with time series of soil moisture, temperature, and climate variables limits our validation to two locations."

In the Results section, we will add the following text (P 17 L 4):

"Figure 9 (c,d NEW) shows observed metabolically active time of biocrusts together with estimates simulated by LiBry for Almería. The observed monthly pattern of active

time is well reproduced by the model (Fig. 9 c), and also for the daily pattern, the model agrees with the measurements (Fig. 9 d). While the response to rain events is captured in general, the model predicts several small to moderate peaks in activity, which do not occur in the observations. Moreover, the model does not entirely reproduce observed periods of prolonged activity in late spring and winter, which leads to a slight underestimation of total annual activity by the model. These findings are discussed below."

[Figure]

**Fig. 9 NEW** Comparison of (a) biocrust water content and (b) surface temperature estimated by LiBry to observations for one field site and one biocrust type (dark cyanobacteria-dominated biocrust) at Soebatsfontein, South Africa. Comparison of (c) monthly and (d) daily biocrust active time fraction simulated by LiBry to observations from a field site near Almería, South Spain.

The Discussion section will be extended as follows (P 21 L 26):

"At the field site of Almería, the simulated monthly and daily patterns of active time match well the observations in general. However, in spring and fall, the model predicts several small and a few larger peaks of activity which cannot be seen in the measurements. One possible explanation for this is that the model may overestimate dew input from the atmosphere, possibly due to the spatially uniform value of maximum dew which is used in LiBry for all dryland regions. In contrast, the model underestimates longer periods of high activity in late spring and winter, which are not directly related to rainfall events (Fig. 9 d NEW). Potential reasons for this may be the missing water storage

capacity of the soil in the model, or underestimation by the model of activation from unsaturated air at relatively high humidity. Although the latter process is represented in the model, it contributes little to the total simulated water supply. Hence, same as for Soebatsfontein, LiBry tends to slightly underestimate active time.

*21. I would consider putting the 1.1 (which is listed as 1.04 in the abstract?) and 0.6 Tg/yr into the Conclusions and adding the 20% in the Abstract.*

The value of 1.1 comes from the paper by Weber et al. [2015], while we estimate 1.04 Tg yr$^{-1}$ NO-N. We agree with the referee and we will add the value of 20% to the abstract and also mention our estimate in numbers (1.04 Tg yr$^{-1}$ NO-N and 0.69 Tg yr$^{-1}$ HONO-N) in the conclusion of the revised manuscript.